# OpenReview forum: "UGround: Towards Unified Visual Grounding with Unrolled Transformers"
_ICLR.cc/2026/Conference — ICLR 2026 Conference Withdrawn Submission_

### Official Review · Reviewer_Bxic · 2025-10-20

**Soundness:** 3
**Presentation:** 3
**Contribution:** 2
**Rating:** 4
**Confidence:** 5

**Summary:**

The paper proposes UGround, a “unified” visual grounding framework that (i) dynamically selects an intermediate transformer layer via a learned policy (“Stochastic Skip Connection”, SSC) and (ii) feeds a similarity map (between the &lt;SEG&gt; token and image tokens) as a soft logit mask into SAM (“Mask as Prompt”, MasP). The method claims benefits over the common recipe that uses the last hidden layer’s &lt;SEG&gt; token as prompt to SAM. Strong results are reported on ReasonSeg, RefCOCO(+/g), and gRefCOCO, with an ablation suggesting synergy between the proposed SSC and MasP.

**Strengths:**

1.  **Good motivation**: The author hypothesizes that the prompt information in the final layer of the model is too high-level, whereas the middle layers may contain more precise, localized cues. This reasoning motivates the exploration of strategies like Mask-as-Prompt and dynamic layer selection. To support this claim, a corresponding analysis is presented in Section 3.2.
2.  **Simple and effective method**: A small layer selector identifies the most optimal intermediate layer for each image-text pair, converting its token-image affinities into a similarity map. This map acts as an additional spatial prompt for the segmenter. The selector is trained with a policy-gradient method, rewarding it for generating prompts that yield accurate masks.
3.  **Strong Performance**: Competitive performance on ReasonSeg, RefCOCO(+/g) and gRefCOCO.
4.  **Adequate Ablation**: Tables 6, 7, 8 show that the proposed modules and design choices are somewhat effective.

**Weaknesses:**

1.  **Unclear Analysis of Dynamic Layer Selection:**
    1.  In the *Why Dynamic Layer Selection* section, the authors compare the full UGround model (presumably including Mask as Prompt) against soft ground truth masks (Figure 2) to conclude that dynamic layer selection is superior. However, UGround contains components *besides* dynamic layer selection. A cleaner experiment, such as a direct comparison between a baseline vs. the baseline + Dynamic Layer Selection, is needed to isolate and validate this specific component's contribution.
2.  **Lack of Justification for Reinforcement Learning (RL):**
    1.  Why can't dynamic layer selection be trained with a simple cross-entropy loss? Given the fixed number of transformer layers, the ground truth could be assigned to the layer producing the largest overlap between its similarity map and the ground truth mask.
    2.  Why not use soft-gating without an explicit loss on the gates?
    The use of RL seems unnecessarily complex. Justification for this choice over simpler alternatives, ideally supported by comparative experiments, is required.
3. **Missing baselines in Table 4**:
    1. CoReS: Orchestrating the Dance of Reasoning and Segmentation, Bao et al., ECCV 2024.
    2. Mask Grounding for Referring Image Segmentation, Chng et al., CVPR 2024.
4.  **Generalization & Limitations:**
    1.  The results focus on datasets like ReasonSeg, RefCOCO, and gRefCOCO. Models trained on these heavily benchmarked datasets may be susceptible to over-fitting. An analysis of the model's limitations and its performance on "in-the-wild" images is needed to demonstrate broader generalization.
5.  **Computational Costs:**
    1.  UGround is expected to have higher training and inference costs due to its "Mask as Prompts" component. A comparison of training and inference costs (e.g., latency, memory usage) against baselines, using the same LLaVA and SAM variants, should be provided.
6.  **Writing and Formatting Issues:**
    1.  *Figure 2:* The legends are too small and difficult to read. Consider increasing the font size for better legibility.
    2.  *Table 2:* There appears to be a typo. $\mathcal{S}\_{\mathrm{IoU}}$ → $\mathcal{M}\_{\mathrm{IoU}}$.
    3.  *Line 261:* There is a capitalization typo. "To this end, We..." should be "To this end, we...".
    4.  *Line 340:* The term "MC Dropout" is mentioned. Please briefly explain this technique and provide a citation.
    5.  *General:* Acronym capitalization is inconsistent (e.g., "LLava" vs. "LLaVA"). Please ensure all acronyms are capitalized uniformly throughout the manuscript.

**Questions:**

1.  Is the comparison in Table 2 fair, given that it compares the original SAM with an adapted SAM?
2.  In Table 6, why does "Mask as Prompt + SSC" (row 5) perform worse than "Mask as Prompt" alone (row 2)?
3.  Is "Unrolled Transformers" an appropriate name for this method? The name is potentially misleading for two reasons: 1) It suggests a new architecture, but the transformer model itself appears to be unchanged. 2) The term "unrolling" does not seem to align with the proposed technique.
4.  Does the dynamic layer selection method choose only one layer? If so, why not select multiple layers, which might improve performance?

---

> ### Author Response · Authors · 2025-11-19
> **Rebuttal by Authors**
>
> **1. Unclear Analysis of Dynamic Layer Selection**
>
> In the upcoming version, we will provide an elaboration of the Analysis in Sec. 3.2 for clarification.
>
> In original work LISA [1],
>
> input: ($x\_{img}$, $x\_{txt}$) $\rightarrow$ LLaVA (layers: $\ell=0,1,..., 31$) $\rightarrow$ **(c)fixed last layer:** $\boldsymbol{h}\_{seg}^{\left( 31 \right)}\in \mathbb{R} ^{4096}$ $\rightarrow$ SAM: $\hat{\mathbf{M}}$
>
> In our UGround, we advance LISA [1] via components **(a)** and **(b)**:
>
> input: ($x\_{img}$, $x\_{txt}$) $\rightarrow$ LLaVA (layers: $\ell=0,1,..., 31$) $\rightarrow$ **(a) layer select**: e.g., $\ell^*$=18 $\rightarrow$ **(b) MasP**:$\mathcal{M}^{(18)}$, $\boldsymbol{h}\_{seg}^{\left( 18 \right)}\in \mathbb{R} ^{4096}$ $\rightarrow$ SAM: $\hat{\mathbf{M}}$
>
> Figure 2  is obtained by  **(b)** vs. **(b) + (a)** setting, a.k.a., baseline vs. baseline + Dynamic Layer Selection. Reviewer presumably prefer the **(c)** vs. **(c)** + **(a)** setting. We believe that either **(b)** or **(c)** justifies its use as the baseline given that:
> 1) **“To analyze the impact of intermediate layers,” as stated in the original paper (Line 201), we focus on analyzing the influence and role of intermediate layers. We aim to investigate why dynamic layer selection improves model performance—its underlying mechanism, rather than quantifying the specific contribution of the dynamic layer selection module itself.** The dynamic layer selection is merely one way to investigate intermediate layers' behaviors. In the introduction (Lines 83–89), we mention that message passing in stacked transformers is analogous to a "phone game." By employing dynamic layer selection, the intermediate layers can directly participate in the interaction and joint training with SAM. As a result, each layer’s performance improves, ultimately leading to an overall improvement in the model. Prior work has not paid attention to the impact of intermediate layers, whereas we leverage dynamic layer selection to explore their effects.
>
> 2) The Analysis in Sec. 3.2 ("Why Dynamic Layer Selection." or "Why Similarity as Mask.") serves to **explain the motivation, thereby leading to the introduction of our method, not to present ablation experiments.**
>
> 3) As for the Dynamic Layer Selection's contribution, it has already been evaluated in Sec.5.3 ablation study (Lines 456-463: Table 6, Exp. 1 vs. Exp. 4, a.k.a. the **(c)** vs. **(c) + (a)** setting, as requested by the reviewer).
>
> 4) The Analysis in Sec. 3.2 is based on relative comparisons, emphasizing relative changes rather than absolute values.
>
> [1] LISA: Reasoning Segmentation via Large Language Model, CVPR 2024.

---

> > ### Author Response · Authors · 2025-11-19
> > **Rebuttal by Authors**
> >
> > **2. Lack of Justification for Reinforcement Learning (RL)**
> >
> > Reinforcement Learning (RL) is introduced to **address the hard-boundary problem** in layer selection, enabling the model to explicitly choose a layer (e.g., layer 26) in a soft, differentiable manner during training. Because hard-boundary layer selection is non-differentiable, **using RL is necessary**.
> >
> > The reviewer raised two questions:  (1) *"Why can't dynamic layer selection be trained with a simple cross-entropy loss?"* and (2) *"Why not use soft-gating without an explicit loss on the gates?"*  As suggested by the reviewer, (1) + (2) yields a simpler alternative, which we refer to as **"CE + soft-gating."**  In this design, the overlap (IoU) between each layer’s similarity map (layers 0–31) and the ground-truth mask is used to assign the layer-selection label. A layer-selection classifier is then trained, and its output scores are used as soft gates to weight and fuse the transformer layers (0–31) into a unified representation, which is subsequently fed into SAM for mask decoding.  However, **CE + soft-gating** suffers from the following issues:
> >
> > 1) **CE + soft-gating is numerically suboptimal.** Consider a scenario after training for 1–2 epochs where the classifier predicts the following scores for layers 0–31: ($[0.01, 0.004, \ldots, 0.7, 0.1, 0.1]$).  Since 0.7 is the maximum score, the 29-th layer is identified as the *best* layer.  However, if we use the full score vector ($[0.01, 0.004, \ldots, 0.7, 0.1, 0.1]$) as soft-gating weights to fuse the embeddings of layers 0–31, the best embedding from layer 29 will inevitably be mixed with features from all other layers (0–28 and 30–31), making it no longer optimal. Soft-gating inherently prevents the remaining layers from receiving zero weight.  In fact, as shown in Figure 6 of the UGround Appendix (and also discussed in [NAACL 2025][1]), transformer layers are activated only at a few specific depths. Assigning non-zero weights to inactive layers dilutes the contribution of these key activated layers.  **Therefore, a hard boundary is required—that is, explicitly selecting a single layer (e.g., assigning 100% weight to layer 29) rather than distributing fractional weights (e.g., 60% on layer 29). This is inherently a discrete decision problem, for which reinforcement learning (RL) provides a principled and trainable solution.**
> >
> > [1] From Redundancy to Relevance: Enhancing Explainability in Multimodal Large Language Models, NAACL 2025.
> >
> > 2) **The training objective of the layer-selection classifier (i.e., CE loss) does not align with the ultimate goal of optimizing the gates through backpropagation from the downstream SAM mask decoding (i.e., use soft-gating without an explicit loss on the gates)**.  The layer-selection classifier explicitly trains the gate to predict which layer is the most important, whereas using soft weights to fuse all layers aims to optimize the overall representation, without explicitly enforcing the gate to select the optimal layer. In fact, during backpropagation, this can interfere with the training of the layer-selection classifier.
> >
> > 3) **Soft-gating may ignore inter-layer dependencies in the transformer**. By weighting and summing all layers, it actually performs a soft ensemble, disrupting the original layer-by-layer structure of the transformer. In addition, it reduces interpretability.
> >
> > 4) **Computational cost: CE + Soft Gating vs. RL** Given $\boldsymbol{h}\_{seg} \in \mathbb{R}^{4096}$ and $\boldsymbol{h}\_{img} \in \mathbb{R}^{576 \times 4096}$, the similarity map is computed as:  $\boldsymbol{h}\_{seg} \times \boldsymbol{h}\_{img}^{T} = (1 \times 4096) \times (576 \times 4096)^{T} = 576 $, followed by reshaping into $24 \times 24$, and then interpolating to $256 \times 256$ to obtain $\mathcal{M}$.  If using the CE + soft-gating approach, one must compute the similarity map for **all 32 layers** of LLaVA, resulting in **32 × 256 × 256** maps, and then compute the overlap (IoU) with the ground-truth mask as the supervision label.  In contrast, with RL, we compute $\boldsymbol{h}\_{seg} \times \boldsymbol{w}^{T} = (1 \times 4096) \times (32 \times 4096)^{T} = 32$, yielding a 32-dimensional **similarity logits** vector, followed by REINFORCE to select the layer index, e.g., ($\ell^* = 28$). $\boldsymbol{w}^T \in \mathbb{R}^{32 \times 4096}$ denotes the layer-gating weights, which are the only learnable parameters in our UGround. The reward $r$ only requires computing the similarity map for **a single layer**. Here we present the core implementation of the **dynamic layer selection module (SSC)**.

---

> ### Author Response · Authors · 2025-11-19
> **Rebuttal by Authors**
>
> ```
> 135    def _policy_walker_mode3(self):
> 136
> 137       # Build per-token layer logits from detached features (condition x) and learnable per-layer weights
> 138       # seg_token_embeds: [N_seg, L, D], layer_gate_W_similarity: [L, D]
> 139       # logits: [N_seg, L]
> 140       logits_similarity = torch.einsum('nld,ld->nl', self.seg_token_embeds.detach(), self._layer_gate_W_similarity)
> 141       # Compute mean layer probabilities across tokens for critic input
> 142       if logits_similarity.size(0) > 0:
> 143               layer_probs = torch.softmax(logits_similarity, dim=-1)  # [N_seg, L]
> 144               self.last_layer_probs_mean = layer_probs.mean(dim=0).detach()  # [L]
> 145       else:
> 146              self.last_layer_probs_mean = None
> 147              return self._policy_walker_mode1()
> 148
> 149       # Per-token sampling from categorical(logits)
> 150       dist = torch.distributions.Categorical(logits=logits_similarity)
> 151       actions = dist.sample()  # [N_seg]
> 152       self.log_probs = dist.log_prob(actions)  # [N_seg]
> 153       batch_indices = torch.arange(self.seg_token_embeds.size(0), device=self.seg_token_embeds.device)
> 154       seg_token_embeds_for_similarity = self.seg_token_embeds[batch_indices, actions]  # [N_seg, D]
> 155       seg_image_token_embeds_for_similarity = self.seg_image_token_embeds[batch_indices, actions]  # [N_seg, P, D]
> 156       # Use the same chosen layer for SAM branch for consistency
> 157       seg_token_embeds_for_sam = seg_token_embeds_for_similarity
> 158       return seg_token_embeds_for_similarity, seg_image_token_embeds_for_similarity, seg_token_embeds_for_sam
> ```
> See also: `UGround/model/PPM.py`
>
> **We want to claim our core contribution**: 1) **Stochastic Skip Connection(SSC)** and 2) **Mask as Prompt(MasP)**, as stated in our paper (**Abstract, Lines 18-26; Sec. 1 introduction, Lines 95-107**). SSC is a reinforcement learning policy that, via stochastic sampling, allows each [SEG] token to slide across unrolled transformer layers, enabling dynamic layer selection at which it connects to the vision model (e.g., SAM) **in a skip-connection fashion ("skip-connection-like'')**[2]. By skip subsequent ones of $\ell^*$, we can ''cheat'' this telephone game (Main text: Lines 82-94) by letting the final participant (SAM) tap into intermediate layers (e.g., 1–39) in advance. Across several forward pass, **such mechanism virtually connects all layers to SAM while activating only one pass at a time ("dropout-like'')**. Given the selected hidden layer, MasP uses the similarity map derived from the [SEG] token and image tokens as a soft logit mask to prompt SAM for mask generation, **offering explicit spatial cues through its activation regions**.
>
> [2] ResNet: Deep Residual Learning for Image Recognition, CVPR 2016.
>
> **3. Missing baselines in Table 4**
>
> We will include these two papers as baselines in Table 4, as requested by the reviewer.
>
> CoReS: Orchestrating the Dance of Reasoning and Segmentation, Bao et al., ECCV 2024.
>
> Mask Grounding for Referring Image Segmentation, Chng et al., CVPR 2024.

---

> ### Author Response · Authors · 2025-11-19
> **Rebuttal by Authors**
>
> **4. Generalization & Limitations**
>
> Currently, **UGround supports 8 dataset types**, namely, A: sem_seg, B: refer_seg, C: neg_refer_seg, D: correct_refer_seg, E: vqa, F: reason_seg, G: reason_seg_plus, and H: multi_reason_seg.
>
> A: sem_seg: ade20k||cocostuff||pascal_part||paco_lvis||mapillary
>
> B: refer_seg: refclef||refcoco||refcoco+||refcocog||refzom||grefcoco
>
> C: neg_refer_seg: R-refcoco||R-refcoco+||R-refcocog
>
> D: correct_refer_seg: fprefcoco||fprefcoco+||fprefcocog
>
> E: vqa: llava_instruct_150k
>
> F: reason_seg: ReasonSeg|train
>
> G: reason_seg_plus(LISA++): instance_seg||cot||conversations||caption
>
> H: multi_reason_seg(muse): MultiReasonSeg|train
>
> Our UGround is trained in a mixed fashion on these 8 types of datasets as needed, rather than being trained only on heavily benchmarked datasets. **It should be noted that all 8 datasets typically consist of "in-the-wild" images (e.g., A: sem_seg)**. As requested by the reviewer, we report results on two newly proposed benchmarks.
>
> **Table 1. Comparisons of the state-of-the-art ''see'' and ''segment'' results on augmented FP-refcoco(+/g) val set**. Numbers of LISA, Cascading, and SESAME are cited from [2]. The see scores measure the binary classification accuracy. The segment scores are (cIoU). ''FP'' is the abbreviation for False Premise, which denotes a query for an object that is absent from the provided image.
>
> Method      | FP-RefCOCO See | FP-RefCOCO Segment | FP-RefCOCO+ See | FP-RefCOCO+ Segment | FP-RefCOCOg See | FP-RefCOCOg Segment
> ----------- | ---------------| ----------------- | ----------------| ------------------ | ----------------| ------------------
> LISA-7B [1]    | 51.36          | 44.00             | 51.32           | 39.62              | 51.25           | 39.64
> Cascading [2]   | 75.59          | 55.18             | 75.03           | 48.64              | 76.07           | 49.98
> SESAME [2]      | 79.84          | 57.93             | 80.00           | 50.81              | 81.78           | 53.79
> UGround (Ours)   | **85.86**      | **62.80**         | **85.10**       | **56.03**          | **86.86**       | **58.55**
>
> [1] LISA: Reasoning Segmentation via Large Language Model, CVPR 2024.
>
> [2] SESAME: See, Say, and Segment: Teaching LMMs to Overcome False Premises, CVPR 2024.
>
> **Table 2. Comparison on MUSE benchmark**. MUSE, a high-quality multi-target reasoning segmentation benchmark.
> | Method                        | w/o SAM | Val gIoU | Val cIoU | Test few targets gIoU | Test few targets cIoU | Test many targets gIoU | Test many targets cIoU | Test overall gIoU | Test overall cIoU |
> |--------------------------------|---------|-----------|-----------|---------------------|---------------------|----------------------|----------------------|-----------------|-----------------|
> | SEEM (lightgray)               | ✓       | 13.6      | 16.2      | 23.6                | 24.9                | 8.5                  | 13.2                 | 11.7            | 15.7            |
> | LISA-7B [1]                      | ✗       | 18.8      | 29.0      | 24.7                | 36.5                | 9.6                  | 24.5                 | 12.8            | 27.1            |
> | LISA-7B_rec [1]                   | ✗      | 24.5      | 31.1      | 30.0                | 30.9                | 12.4                 | 23.2                 | 16.2            | 24.8            |
> | LISA-7B_aug [1]                   | ✗      | 42.0      | 46.1      | 43.5                | 52.0                | 37.7                 | 42.3                 | 38.9            | 44.4            |
> | PixelLM-7B  [2]                    | ✓       | 42.6      | 50.7      | 44.6                | 59.2                | 37.7                 | 42.8                 | 39.2            | 46.3            |
> | UGround-7B (Ours)                      | ✓       | **52.4**      | **56.4**      | **46.2**                | **69.7**                | **38.8**                 | **46.9**                 | **40.1**            | **50.4**            |
>
> [1] LISA: Reasoning Segmentation via Large Language Model, CVPR 2024.
>
> [2] PixelLM: PixelLM: Pixel Reasoning with Large Multimodal Model, CVPR 2024.

---

> > ### Author Response · Authors · 2025-11-19
> > **Rebuttal by Authors**
> >
> > **5. Computational Costs**
> >
> > **The training and inference costs of UGround are very low and can be considered negligible**.
> >
> > We report UGround’s training and inference costs on a single NVIDIA A100-SXM4-40GB GPU. We train on the ReasonSeg train split over 400 samples for 1 epoch (200 steps), with a batch size of 2. *Training Latency (s)* refers to the average time per iteration during the 200-step training process (including forward, backward, and optimizer step), and *Memory Usage* refers to the MaxMemAllocated observed within the 200 steps. We also report the model’s *Trainable Percentage* (Trainable Parameters / Total Parameters). For inference, we evaluate on the ReasonSeg test split over 799 samples, with a batch size of 1. *Speed (FPS)* refers to the average frames per second computed over the 799 samples.
> >
> > **Comparing the training cost of our UGround to state-of-the-art methods**.
> > | Model   | Batch Size | Training Latency (s) | Memory Usage (GB) | Trainable Percentage | Trainable Parameters | Total Parameters |
> > | :-------  | :------------: | :------------: | :----------:  | :--------------------: | :--------------------: | :----------------: |
> > | LISA [3]    | 2 |1.26         | 23.68      | 3.74%                | 288.25M (288,251,364)          | 7.71B (7,708,933,424)    |
> > | SESAME [2] | 2 | 1.11         | 23.15      | 3.73%                | 288.25M (288,251,364)          | 7.73 B (7,725,714,736)    |
> > | GSVA [1]   | 2         | 2.13      |   25.73        | 3.73%                | 288.26M (288,259,556)          | 7.73B (7,726,050,608)    |
> > | PixelLM [4] | 2         | 1.22      |   23.02       | 5.25%                | 375.72M (375,724,772)          | 7.16B (7,157,608,240)    |
> > | UGround (Ours) | 2 | 2.97         | 28.16      | 3.72%                | 288.40M (288,400,944)          | 7.75 B (7,747,165,488)    |
> >
> > **Comparing the runtime speed of our UGround to state-of-the-art methods**.
> > | Model        | GSVA [1] | SESAME [2] | LISA [3] |PixelLM [4]|UGround|
> > |:------------:|:--------:|:----------:|:-----------:|:-----------:|:-----------:|
> > | Speed (FPS)  |   3.98   |    4.64    |   4.68      |    9.24     |    4.12     |
> >
> > [1] GSVA: Generalized Segmentation via Multimodal Large Language Models, CVPR 2024.
> >
> > [2] SESAME: See, Say, and Segment: Teaching LMMs to Overcome False Premises, CVPR 2024.
> >
> > [3] LISA: Reasoning Segmentation via Large Language Model, CVPR 2024.
> >
> > [4] PixelLM: Pixel Reasoning with Large Multimodal Model, CVPR 2024.
> >
> > **In fact, we only use the most basic reinforcement learning formulation, REINFORCE, rather than more advanced methods such as PPO, or GRPO. The only additional trainable parameters we introduce are the 32×4096 parameters (PPM.py: Lines 32–34)**.
> >
> > See also: `UGround/model/PPM.py`
> > ```
> >   31      # Layer gating weights (initialized with provided hidden_dim)
> >   32      self._layer_gate_W_similarity = nn.Parameter(
> >   33                 torch.ones(self.num_layers, self.hidden_dim)
> >   34      )   # self.num_layers=32, self.hidden_dim=4096
> > ```
> >
> > We understand the reviewer’s concerns regarding the costs of Mask as Prompt, and we provide a more detailed explanation here.
> >
> > We use SAM [1] for mask decoding. In the original paper, SAM states:
> >
> > > **Prompt encoder.** We consider two sets of prompts: sparse (points, boxes, text) and **dense (masks)**. We represent points and boxes by positional encodings [93] summed with learned embeddings for each prompt type, and free-form text with an off-the-shelf text encoder from CLIP [80]. **Dense prompts (i.e., masks) are embedded using convolutions and summed element-wise with the image embedding.**
> >
> > That is, SAM takes a mask as a prompt. The SAM mask prompt is a 256×256 input, and after passing through the convolutional layers, its dimensions change as follows:
> > | Layer                          | Output shape                          |
> > |---------------------------------|--------------------------------|
> > | Input mask                      | (B, 1, 256, 256)               |
> > | Conv2d(1 → 4, stride=2)         | (B, 4, 128, 128)               |
> > | Conv2d(4 → 16, stride=2)        | (B, 16, 64, 64)                |
> > | Conv2d(16 → 256, stride=1)      | (B, 256, 64, 64)               |
> >
> > So, the final dimension of the mask embedding is **[B, 256, 64, 64]**. In SAM’s implementation, even when the mask-as-prompt input is not used — **that is, when only the text [SEG] prompt is provided and `mask is None` — SAM still generates a set of learnable embeddings with shape [B, 256, 64, 64]**(`prompt_encoder.py`: Line 65, Line 182), which serve as the mask-embedding representation.  **Therefore, Mask as Prompt does not introduce any additional parameter overhead.**
> >
> > See also: `UGround/model/segment_anything/modeling/prompt_encoder.py`

---

> ### Author Response · Authors · 2025-11-19
> **Rebuttal by Authors**
>
> ```
>    56  self.mask_downscaling = nn.Sequential(      # mask convolutions
>    57     nn.Conv2d(1, mask_in_chans // 4, kernel_size=2, stride=2),   # mask_in_chans=16
>    58     LayerNorm2d(mask_in_chans // 4),
>    59     activation(),
>    60     nn.Conv2d(mask_in_chans // 4, mask_in_chans, kernel_size=2, stride=2),
>    61     LayerNorm2d(mask_in_chans),
>    62     activation(),
>    63     nn.Conv2d(mask_in_chans, embed_dim, kernel_size=1),    # embed_dim=256
>    64     )
>    65     self.no_mask_embed = nn.Embedding(1, embed_dim)  # embed_dim=256
>    ...
>   111    def _embed_masks(self, masks: torch.Tensor) -> torch.Tensor:
>   112         """Embeds mask inputs."""
>   113         mask_embedding = self.mask_downscaling(masks)  # masks=[B, 1, 256, 256]
>   114         return mask_embedding   # mask_embedding=[B, 256, 64, 64]
>    ...
>   140    def forward(
>   141          self,
>   142          points: Optional[Tuple[torch.Tensor, torch.Tensor]],
>   143          boxes: Optional[torch.Tensor],
>   144          masks: Optional[torch.Tensor],
>   145          text_embeds: Optional[torch.Tensor],
>   146     ) -> Tuple[torch.Tensor, torch.Tensor]:
>    ...
>    176     if text_embeds is not None:
>    177         sparse_embeddings = torch.cat([sparse_embeddings, text_embeds], dim=1) #sparse_embeddings=[B, 1, 256]
>    178
>    179     if masks is not None:
>    180           dense_embeddings = self._embed_masks(masks) # dense_embeddings=[B, 256, 64, 64]
>    181     else:
>    182           dense_embeddings = self.no_mask_embed.weight.reshape(1, -1, 1, 1).expand(
>    183                 bs, -1, self.image_embedding_size[0], self.image_embedding_size[1]
>    184           )   # dense_embeddings=[B, 256, 64, 64]
>    185     return sparse_embeddings, dense_embeddings
> ```
>
> We have included the code in the supplementary materials, which you can download here: **[supplementary_material](https://openreview.net/attachment?id=pWi3tvhhmx&name=supplementary_material)**.
>
> [1] SAM: Segment Anything, ICCV 2023.
>
> **6. Writing and Formatting Issues**
> >1. Figure 2: The legends are too small and difficult to read. Consider increasing the font size for better legibility.
> >2. Table 2: There appears to be a typo. $\mathcal{S} _{\mathrm{IoU}}\rightarrow \mathcal{M} _{\mathrm{IoU}}$
> >3. Line 261: There is a capitalization typo. "To this end, We..." should be "To this end, we...".
>
> We have fixed it in the revision.
>
> >Line 340: The term "MC Dropout" is mentioned. Please briefly explain this technique and provide a citation.
>
> We will provide a more detailed explanation in the revised version once the one-page extension is approved. Also, we will provide a citation to Gal and Ghahramani [1], *Dropout as a Bayesian Approximation: Representing Model Uncertainty in Deep Learning*, ICML 2016.
>
> [1] Dropout as a bayesian approximation: Representing model uncertainty in deep learning, ICML 2016.
>
> Here, we provide an example to illustrate "MC Dropout" for the reviewer.
>
> Consider the query: *“Can you segment the apple, orange, and banana in the picture?”*
> UGround responds: *“Yes, apple, orange, and banana are [seg], [seg], [seg].”* That is, there are three objects to segment in a single image. During one forward pass, our SSC module (RL) selects a layer from 0–31 for each of $[seg]\_{apple}$, $[seg]\_{orange}$, and $[seg]\_{banana}$ as input to SAM. For example, suppose layers (15, 28, 19) are selected, i.e., $[seg]\_{apple}^{(15)} \rightarrow$ SAM, $[seg]\_{orange}^{(28)} \rightarrow$ SAM, $[seg]_{banana}^{(19)} \rightarrow$ SAM.
>
> In the $(n-1)$-th forward pass, the selections are $[seg]\_{apple}^{(16)} \rightarrow$ SAM, $[seg]\_{orange}^{(20)} \rightarrow$ SAM, $[seg]\_{banana}^{(14)} \rightarrow$ SAM.
>
> In the $n$-th forward pass, the selections are $[seg]\_{apple}^{(13)} \rightarrow$ SAM, $[seg]\_{orange}^{(17)} \rightarrow$ SAM, $[seg]\_{banana}^{(26)} \rightarrow$ SAM.
>
> During the $n$ forward passes, $[seg]\_{apple}$ from layers 15, 16, and 13 are independently connected to SAM, and the SSC module (RL) dynamically alters the model’s internal connectivity. This can be analogized to the dropout mechanism in neural networks:  All layers are virtually connected to SAM, but only one path is activated per forward pass. Since the SSC module performs stochastic sampling, this can be interpreted as a Monte Carlo uncertainty estimation. **Unlike soft-gating, the SSC module enables each layer to connect independently to SAM, resulting in 32 independent subnetworks in an ensemble-style design. As only one layer is activated at a time, the computational overhead is comparable to selecting a fixed last layer, thus avoiding expensive computational costs.**
> >5. General: Acronym capitalization is inconsistent (e.g., "LLava" vs. "LLaVA"). Please ensure all acronyms are capitalized uniformly throughout the manuscript.
>
> We have fixed it in the revision.

---

> > ### Author Response · Authors · 2025-11-19
> > **Rebuttal by Authors**
> >
> > **Questions**
> > >1. Is the comparison in Table 2 fair, given that it compares the original SAM with an adapted SAM?
> >
> > "An adapted SAM" — here, "adapted" means that the SAM model can receive text (i.e., [seg] token) as a prompt. In the original SAM paper, it does not accept text as a prompt; **LISA [1] modifies it. LISA's successors, such as SESAME [2], GSVA [3], and PixelLM [4], all adopt this paradigm**.
> > ```
> > 140. def forward(
> >               self,
> >               points: Optional[Tuple[torch.Tensor, torch.Tensor]],
> >               boxes: Optional[torch.Tensor],
> >               masks: Optional[torch.Tensor],
> >               text_embeds: Optional[torch.Tensor],   # LISA [1] modifies it here !
> >         ) -> Tuple[torch.Tensor, torch.Tensor]:
> > ...
> >         if text_embeds is not None:   # LISA [1] modifies it here !
> >             sparse_embeddings = torch.cat([sparse_embeddings, text_embeds], dim=1)
> >         return sparse_embeddings, dense_embeddings
> > ```
> > See also: `UGround/model/segment_anything/modeling/prompt_encoder.py`
> >
> > The Analysis in Sec. 3.2 ("Why Similarity as Mask.") serves to **explain the motivation, thereby leading to the introduction of our method, not to present ablation experiments.**  LISA and its successors, such as SESAME, GSVA, and PixelLM, all use [seg] as a prompt, which has been demonstrated to be feasible. In contrast, we aim to use the logit mask (similarity map) as a prompt, and **we want to investigate whether similarity map is feasible**. To this end, we implemented the following: to measure the consistency between the similarity map and the ground truth mask, we compute the $\mathcal{M}\_{\mathrm{IoU}}$ between them; to probe whether SAM can understand the semantics of the similarity map, we directly use the similarity map to prompt the original SAM (without training), denoted as $\mathcal{M}\_{\mathrm{prompt}}$.
> >
> > [1] LISA: Reasoning Segmentation via Large Language Model, CVPR 2024.
> >
> > [2] SESAME: See, Say, and Segment: Teaching LMMs to Overcome False Premises, CVPR 2024.
> >
> > [3] GSVA: Generalized Segmentation via Multimodal Large Language Models, CVPR 2024.
> >
> > [4] PixelLM: Pixel Reasoning with Large Multimodal Model, CVPR 2024.
> >
> > >2. In Table 6, why does "Mask as Prompt + SSC" (row 5) perform worse than "Mask as Prompt" alone (row 2)?
> >
> > For simplicity, we train [seg] as prompt and mask as prompt together, and then break them down for analysis. Exp. IDs 1–3 form one group of experiments, and Exp. IDs 4–6 form another group. The fact that Exp. ID 2 (row 2) outperforms Exp. ID 5 (row 5) is because [seg] as prompt takes a share of the contribution.

---

> ### Author Response · Authors · 2025-11-19
> **Rebuttal by Authors**
>
> >3. Is "Unrolled Transformers" an appropriate name for this method? The name is potentially misleading for two reasons: 1) It suggests a new architecture, but the transformer model itself appears to be unchanged. 2) The term "unrolling" does not seem to align with the proposed technique.
>
> LLaVA consists of 32 Transformer layers stacked in a loop, with each layer having a dimension of `batch_size × seq_len × hidden_dim`. We collect the output tensor from each layer and stack them into a tensor array of shape: `batch_size × layer_num × seq_len × hidden_dim`, **which is equivalent to "unrolling" the tensors of all layers into an array**, where layer_num = 32. Our SSC (RL) dynamically selects layers, **which can be viewed as sliding across the unrolled layers (Transformers) according to the policy prediction**.
> ```
> 54   def recurrent_unrolled(self, hidden_states, input_ids, seg_token_mask, num_patches):
> 55         # hidden_states: [B, L, T, D],   L: layer_num,   T: seq_len
> 56         hidden_states = torch.stack(hidden_states,dim=1)
> 57         B, L, T, D = hidden_states.shape
> 58         device = hidden_states.device
> 59
> 60         image_token_mask = (input_ids == IMAGE_TOKEN_INDEX)         # [B, T]
> 61         image_token_idx = image_token_mask.float().masked_fill(~image_token_mask, float('inf')).argmin(dim=1)  # [B]
> 62
> 63         idx_offset = torch.arange(num_patches, device=device).unsqueeze(0).expand(B, -1)  # [B, num_patches]
> 64         gather_idx = image_token_idx.unsqueeze(1) + idx_offset  # [B, num_patches]
> 65         gather_idx = gather_idx.unsqueeze(1).unsqueeze(-1).expand(-1, L, -1, D)  # [B, L, num_patches, D]
> 66
> 67         seg_token_mask_bool = seg_token_mask.bool()  # [B, T]
> 68         all_batch_indices = torch.arange(B, device=device).unsqueeze(1).expand_as(seg_token_mask)  # [B, T]
> 69         batch_idx = all_batch_indices[seg_token_mask_bool]  # [N_seg]
> 70         # token_idx = torch.arange(T, device=device).unsqueeze(0).expand(B, -1)[seg_token_mask_bool]  # [N_seg]
> 71
> 72         seg_token_embeds = hidden_states.permute(0, 2, 1, 3)  # [B, T, L, D]
> 73         seg_token_embeds = seg_token_embeds[seg_token_mask_bool]  # [N_seg, L, D]
> 74
> 75         image_token_embeds = torch.gather(hidden_states, dim=2, index=gather_idx)  # [B, L, num_patches, D]
> 76         seg_image_token_embeds = image_token_embeds[batch_idx]  # [N_seg, L, num_patches, D]
> 77
> 78         return seg_token_embeds, seg_image_token_embeds
> ```
> See also: `UGround/model/PPM.py`
>
> >4. Does the dynamic layer selection method choose only one layer? If so, why not select multiple layers, which might improve performance?
>
> Yes, the dynamic layer selection method chooses only one layer **for each [seg] token in a single forward pass**. However, for a multi-target task, a sample may contain multiple objects, and the dynamic layer selection will choose a layer for each [seg] token, **depending on the number of [seg] tokens in the sample**.
>
> For simplicity, here we consider only the **single-object segmentation** case, in which a sample contains only one [seg] token.
> Suppose the dynamic layer selection (a.k.a. SSC, RL policy) in UGround,
>
> chooses **the 18th layer** during the **1st** forward pass:
>
> input: ($x\_{img}$, $x\_{txt}$) $\rightarrow$ LLaVA (layers: $\ell=0,1,..., 31$) $\rightarrow$ **(a) layer select**: e.g., $\ell^*$=18 $\rightarrow$ **(b) MasP**:$\mathcal{M}^{(18)}$, $\boldsymbol{h}\_{seg}^{\left( 18 \right)}\in \mathbb{R} ^{4096}$ $\rightarrow$ SAM: $\hat{\mathbf{M}}$
>
> chooses **the 20th layer** during the **$(n-1)$-th** forward pass:
>
> input: ($x\_{img}$, $x\_{txt}$) $\rightarrow$ LLaVA (layers: $\ell=0,1,..., 31$) $\rightarrow$ **(a) layer select**: e.g., $\ell^*$=20 $\rightarrow$ **(b) MasP**:$\mathcal{M}^{(20)}$, $\boldsymbol{h}\_{seg}^{\left( 20 \right)}\in \mathbb{R} ^{4096}$ $\rightarrow$ SAM: $\hat{\mathbf{M}}$
>
> chooses **the 29th layer** during the **$(n)$-th** forward pass:
>
> input: ($x\_{img}$, $x\_{txt}$) $\rightarrow$ LLaVA (layers: $\ell=0,1,..., 31$) $\rightarrow$ **(a) layer select**: e.g., $\ell^*$=29 $\rightarrow$ **(b) MasP**:$\mathcal{M}^{(29)}$, $\boldsymbol{h}\_{seg}^{\left( 29 \right)}\in \mathbb{R} ^{4096}$ $\rightarrow$ SAM: $\hat{\mathbf{M}}$
>
> As $n$ gradually increases, i.e., as the number of steps grows, **the model is expected to eventually converge to the optimal layer for each [seg] token**. For a single [seg] token, selecting multiple layers in a single operation requires fusing the features from these layers, which may result in inactive layers diluting the contributions of active layers (This issue has already been discussed in *Weaknesses: 2. Lack of Justification for Reinforcement Learning (RL)*).

---

> > ### Author Response · Authors · 2025-11-19
> > **Official Comment by Authors**
> >
> > Dear reviewer, we would like to thank you for your time and efforts. We hope that your concerns are addressed with our rebuttal. Please let us know if there are any further questions that need clarification.

---

> > > ### Comment · Reviewer_Bxic · 2025-11-23
> > > **Has corresponding changes been made to the main paper?**
> > >
> > > I have not seen any changes made to the original paper. If the revised paper has not yet been uploaded, please do so and explicitly mark the modifications. I will only reconsider the score based on the changes incorporated into the main paper itself.

---

> ### Author Response · Authors · 2025-11-26
> **The revised paper is now available.**
>
> Hi Reviewer Bxic, the revised paper is now available at [https://openreview.net/pdf?id=pWi3tvhhmx](https://openreview.net/pdf?id=pWi3tvhhmx). All modifications are clearly highlighted in **blue** throughout the manuscript.
>
> * **Justification for Reinforcement Learning (RL)** is supported by comparative experiments (Lines 480–485) and further discussed in Lines 726–764.
>
> * **Missing baselines** have been added in Table 4 (Lines 408 and 412).
>
> * **Generalization**: Two newly proposed benchmarks are reported in Lines 525–532.
>   **Limitations**: Lines 864–880.
>
> * **Computational costs** are reported in Lines 514–524.
>
> * **Writing and formatting improvements**:
>   * Figure 2 has been revised for better readability (Lines 198–210)..
>   * The typo in Table 2 has been fixed, and the associated section (“Why Similarity as Mask”) has been rewritten (Lines 219–234).
>   * The text related to “MC Dropout” has been rewritten and further clarified with a citation (Lines 324–342 and 791–809).
>   * All acronyms are now capitalized consistently throughout the manuscript. The main paper has been carefully checked and polished.

---

### Official Review · Reviewer_L9cY · 2025-10-31

**Soundness:** 4
**Presentation:** 4
**Contribution:** 4
**Rating:** 8
**Confidence:** 4

**Summary:**

This paper introduces UGround, a novel paradigm for unified visual grounding that aims to create a single model capable of handling diverse grounding tasks, including reasoning-based, multi-target, and false-premise queries. The authors identify the limitations of existing methods that rely on the fixed final hidden layer of a transformer. Their proposed solution, Policy-Prompted Masking (PPM), uses a reinforcement learning policy to dynamically select an optimal intermediate transformer layer. The similarity map from this selected layer is then used as an explicit spatial prompt for a vision decoder (SAM), while also being directly supervised. The proposed framework achieves new SOTA results on several challenging benchmarks.

**Strengths:**

- The core idea of "unrolling" transformers and using a learned policy to dynamically select an intermediate layer is highly novel. This reframes feature extraction from a fixed pipeline to a dynamic, content-aware process. The formulation of layer selection as a reinforcement learning task is a novel and effective approach.
- The paper is supported by strong and comprehensive empirical evidence. The method significantly outperforms recent SOTA methods on diverse and challenging datasets (ReasonSeg, gRefCOCO). The ablation studies are thorough, validating the contribution of each component of the proposed PPM mechanism (dynamic selection, mask as prompt, reward formulation), which substantiates the design choices.
- The paper is very well-written and presented. The motivation is clearly articulated through insightful analysis, and the proposed method is explained with clarity and well-designed figures (Figure 1, Figure 3).
- The work makes a significant contribution by successfully unifying multiple complex visual grounding tasks within a single, coherent framework.
- Honorable mention to the attention given to details (e.g., cleaned and underlined proceedings names in the reference, high-quality figures …) **reflects a high degree of care**.

**Weaknesses:**

While I think the paper is already great, the following points could benefit from clarification to improve the impact of the paper:

- **Inference Overhead and Ambiguity:** The paper does not sufficiently detail the inference-time procedure and its associated costs. Storing intermediate activations from all layers to feed the policy network may introduce significant memory and computational overhead.
- **Limited Conditioning of the Selection Policy:** The layer-selection policy `π(l | H_t*)` is conditioned only on the hidden states of the `<SEG>` token. The optimal layer for feature extraction could plausibly depend on the visual complexity of the image itself. Conditioning the policy on image token representations as well might lead to a more robust and adaptive selection mechanism.
- **Complexity of RL-based Solution:** While effective, the use of a REINFORCE-based policy introduces significant complexity to the training pipeline compared to end-to-end differentiable alternatives. The paper would be strengthened by a justification or comparison against simpler, non-RL methods for layer selection, such as a learned soft-attention mechanism or a weighted average over layer outputs.

**Questions:**

1. What is the practical impact on inference-time latency and memory usage compared to the baseline model?
2. The layer selection policy is conditioned solely on the textual `<SEG>` token's representations. Have the authors considered or experimented with also conditioning the policy on visual information (e.g., a pooled representation of image tokens)?
3. The reward `r` is derived from the similarity map's alignment with the ground-truth mask. Have the authors explored alternative reward signals, such as the final segmentation IoU score generated by SAM? Such a reward might more directly optimize for the final task performance.

---

> ### Author Response · Authors · 2025-11-19
> **Rebuttal by Authors**
>
> **1. Inference Overhead and Ambiguity**
>
> The training and inference costs of UGround are very low and can be considered negligible.
>
> We report UGround’s training and inference costs on a single NVIDIA A100-SXM4-40GB GPU. We train on the ReasonSeg train split over 400 samples for 1 epoch (200 steps), with a batch size of 2. *Training Latency (s)* refers to the average time per iteration during the 200-step training process (including forward, backward, and optimizer step), and *Memory Usage* refers to the MaxMemAllocated observed within the 200 steps. We also report the model’s *Trainable Percentage* (Trainable Parameters / Total Parameters). For inference, we evaluate on the ReasonSeg test split over 799 samples, with a batch size of 1. *Speed (FPS)* refers to the average frames per second computed over the 799 samples.
>
> **Comparing the training cost of our UGround to state-of-the-art methods**.
> | Model   | Batch Size | Training Latency (s) | Memory Usage (GB) | Trainable Percentage | Trainable Parameters | Total Parameters |
> | :-------  | :------------: | :------------: | :----------:  | :--------------------: | :--------------------: | :----------------: |
> | LISA [3]    | 2 |1.26         | 23.68      | 3.74%                | 288.25M (288,251,364)          | 7.71B (7,708,933,424)    |
> | SESAME [2] | 2 | 1.11         | 23.15      | 3.73%                | 288.25M (288,251,364)          | 7.73 B (7,725,714,736)    |
> | GSVA [1]   | 2         | 2.13      |   25.73        | 3.73%                | 288.26M (288,259,556)          | 7.73B (7,726,050,608)    |
> | PixelLM [4]| 2         | 1.22      |   23.02       | 5.25%                | 375.72M (375,724,772)          | 7.16B (7,157,608,240)    |
> | UGround (Ours)| 2 | 2.97         | 28.16      | 3.72%                | 288.40M (288,400,944)          | 7.75 B (7,747,165,488)    |
>
> **Comparing the runtime speed of our UGround to state-of-the-art methods**.
> | Model        | GSVA [1] | SESAME [2] | LISA [3] |PixelLM [4]|UGround|
> |:------------:|:--------:|:----------:|:-----------:|:-----------:|:-----------:|
> | Speed (FPS)  |   3.98   |    4.64    |   4.68      |    9.24     |    4.12     |
>
> [1] GSVA: Generalized Segmentation via Multimodal Large Language Models, CVPR 2024.
>
> [2] SESAME: See, Say, and Segment: Teaching LMMs to Overcome False Premises, CVPR 2024.
>
> [3] LISA: Reasoning Segmentation via Large Language Model, CVPR 2024.
>
> [4] PixelLM: Pixel Reasoning with Large Multimodal Model, CVPR 2024.
>
> We understand the reviewer’s concerns regarding the computational overhead, and we provide a more detailed explanation here.
>
> For simplicity, here we consider only the **single-object segmentation** case, in which a sample contains only one [seg] token.
> Suppose the dynamic layer selection (a.k.a. SSC, RL policy) in UGround,
>
> chooses **the 18th layer** during the **1st** forward pass:
>
> input: ($x\_{img}$, $x\_{txt}$) $\rightarrow$ LLaVA (layers: $\ell=0,1,..., 31$) $\rightarrow$ **(a) layer select**: e.g., $\ell^*$=18 $\rightarrow$ **(b) MasP**:$\mathcal{M}^{(18)}$, $\boldsymbol{h}\_{seg}^{\left( 18 \right)}\in \mathbb{R} ^{4096}$ $\rightarrow$ SAM: $\hat{\mathbf{M}}$
>
> In **(a) layer select**, we first determine the selected layer $\ell^{\*}$, and then retrieve the corresponding text embedding and image embedding from that layer. Therefore, within a single forward pass, we **only need to compute a single similarity map** $\mathcal{M}^{(18)}$ (a.k.a., activations) for the selected layer $\ell^{\*}$. Here we present the core implementation of the dynamic layer selection module (SSC).

---

> > ### Author Response · Authors · 2025-11-19
> > **Rebuttal by Authors**
> >
> > ```
> > 135    def _policy_walker_mode3(self):
> > 136
> > 137       # Build per-token layer logits from detached features (condition x) and learnable per-layer weights
> > 138       # seg_token_embeds: [N_seg, L, D], layer_gate_W_similarity: [L, D]
> > 139       # logits: [N_seg, L]
> > 140       logits_similarity = torch.einsum('nld,ld->nl', self.seg_token_embeds.detach(), self._layer_gate_W_similarity)
> > 141       # Compute mean layer probabilities across tokens for critic input
> > 142       if logits_similarity.size(0) > 0:
> > 143               layer_probs = torch.softmax(logits_similarity, dim=-1)  # [N_seg, L]
> > 144               self.last_layer_probs_mean = layer_probs.mean(dim=0).detach()  # [L]
> > 145       else:
> > 146              self.last_layer_probs_mean = None
> > 147              return self._policy_walker_mode1()
> > 148
> > 149       # Per-token sampling from categorical(logits)
> > 150       dist = torch.distributions.Categorical(logits=logits_similarity)
> > 151       actions = dist.sample()  # [N_seg]
> > 152       self.log_probs = dist.log_prob(actions)  # [N_seg]
> > 153       batch_indices = torch.arange(self.seg_token_embeds.size(0), device=self.seg_token_embeds.device)
> > 154       seg_token_embeds_for_similarity = self.seg_token_embeds[batch_indices, actions]  # [N_seg, D]
> > 155       seg_image_token_embeds_for_similarity = self.seg_image_token_embeds[batch_indices, actions]  # [N_seg, P, D]
> > 156       # Use the same chosen layer for SAM branch for consistency
> > 157       seg_token_embeds_for_sam = seg_token_embeds_for_similarity
> > 158       return seg_token_embeds_for_similarity, seg_image_token_embeds_for_similarity, seg_token_embeds_for_sam
> > ```
> > See also: `UGround/model/PPM.py`
> >
> > We do need to store the hidden states from layers 0–31, with a shape of `batch_size × layer_num × seq_len × hidden_dim`. However, these hidden states from layers 0–31 are already stored by the language model itself during autoregressive generation, and then discarded once text prediction is finished. We simply delay the release of these hidden states by a few steps. From the hidden states of layers 0–31, we extract the text embeddings with shape `batch_size × layer_num × hidden_dim` (e.g., `3 × 32 × 4096`), and the image embeddings with shape `batch_size × layer_num × img_len × hidden_dim` (e.g., `3 × 32 × 576 × 4096`). **All operations are performed on CUDA, and the processing time is only 0.0192 seconds**. Once everything is prepared, we only need to wait for the selection of the target layer $\ell^{*}$.
> >
> > ```
> > 54   def recurrent_unrolled(self, hidden_states, input_ids, seg_token_mask, num_patches):
> > 55         # hidden_states: [B, L, T, D],   L: layer_num,   T: seq_len
> > 56         hidden_states = torch.stack(hidden_states,dim=1)
> > 57         B, L, T, D = hidden_states.shape
> > 58         device = hidden_states.device
> > 59
> > 60         image_token_mask = (input_ids == IMAGE_TOKEN_INDEX)         # [B, T]
> > 61         image_token_idx = image_token_mask.float().masked_fill(~image_token_mask, float('inf')).argmin(dim=1)  # [B]
> > 62
> > 63         idx_offset = torch.arange(num_patches, device=device).unsqueeze(0).expand(B, -1)  # [B, num_patches]
> > 64         gather_idx = image_token_idx.unsqueeze(1) + idx_offset  # [B, num_patches]
> > 65         gather_idx = gather_idx.unsqueeze(1).unsqueeze(-1).expand(-1, L, -1, D)  # [B, L, num_patches, D]
> > 66
> > 67         seg_token_mask_bool = seg_token_mask.bool()  # [B, T]
> > 68         all_batch_indices = torch.arange(B, device=device).unsqueeze(1).expand_as(seg_token_mask)  # [B, T]
> > 69         batch_idx = all_batch_indices[seg_token_mask_bool]  # [N_seg]
> > 70         # token_idx = torch.arange(T, device=device).unsqueeze(0).expand(B, -1)[seg_token_mask_bool]  # [N_seg]
> > 71
> > 72         seg_token_embeds = hidden_states.permute(0, 2, 1, 3)  # [B, T, L, D]
> > 73         seg_token_embeds = seg_token_embeds[seg_token_mask_bool]  # [N_seg, L, D]
> > 74
> > 75         image_token_embeds = torch.gather(hidden_states, dim=2, index=gather_idx)  # [B, L, num_patches, D]
> > 76         seg_image_token_embeds = image_token_embeds[batch_idx]  # [N_seg, L, num_patches, D]
> > 77
> > 78         return seg_token_embeds, seg_image_token_embeds
> > ```
> > See also: `UGround/model/PPM.py`

---

> > > ### Author Response · Authors · 2025-11-19
> > > **Rebuttal by Authors**
> > >
> > > **2. Limited Conditioning of the Selection Policy**
> > >
> > > We agree.  Currently, the layer-selection policy $\pi_\theta(\ell \mid \mathcal{H}_{t^{*}})$ is conditioned only on the hidden states of the [SEG] token. In a future journal extension, we plan to advance UGround by conditioning the policy on image token representations as well. It is **important to note that image token representations can serve only as *auxiliary* information (Indeed, the image itself could plausibly provide more informative cues for layer selection.)—they cannot condition the policy on their own**. In a multi-target setting, where a single sample contains multiple [SEG] tokens, we need to select a dedicated layer for each [SEG] token. In this case, the image token representations act as shared information across targets: they cannot independently determine the layer for each [SEG] token, but they may help provide a more robust and adaptive selection mechanism when used as supplementary features.  In our implementation, incorporating image token representations into the policy conditioning is straightforward (**Line 140**).
> > > ```
> > > 135    def _policy_walker_mode3(self):
> > > 136
> > > 137       # Build per-token layer logits from detached features (condition x) and learnable per-layer weights
> > > 138       # seg_token_embeds: [N_seg, L, D], layer_gate_W_similarity: [L, D]
> > > 139       # logits: [N_seg, L]
> > > 140       logits_similarity = torch.einsum('nld,ld->nl', self.seg_token_embeds.detach(), self._layer_gate_W_similarity)
> > > 141       # Compute mean layer probabilities across tokens for critic input
> > > 142       if logits_similarity.size(0) > 0:
> > > 143               layer_probs = torch.softmax(logits_similarity, dim=-1)  # [N_seg, L]
> > > 144               self.last_layer_probs_mean = layer_probs.mean(dim=0).detach()  # [L]
> > > 145       else:
> > > 146              self.last_layer_probs_mean = None
> > > 147              return self._policy_walker_mode1()
> > > 148
> > > 149       # Per-token sampling from categorical(logits)
> > > 150       dist = torch.distributions.Categorical(logits=logits_similarity)
> > > 151       actions = dist.sample()  # [N_seg]
> > > 152       self.log_probs = dist.log_prob(actions)  # [N_seg]
> > > 153       batch_indices = torch.arange(self.seg_token_embeds.size(0), device=self.seg_token_embeds.device)
> > > 154       seg_token_embeds_for_similarity = self.seg_token_embeds[batch_indices, actions]  # [N_seg, D]
> > > 155       seg_image_token_embeds_for_similarity = self.seg_image_token_embeds[batch_indices, actions]  # [N_seg, P, D]
> > > 156       # Use the same chosen layer for SAM branch for consistency
> > > 157       seg_token_embeds_for_sam = seg_token_embeds_for_similarity
> > > 158       return seg_token_embeds_for_similarity, seg_image_token_embeds_for_similarity, seg_token_embeds_for_sam
> > > ```
> > > See also: `UGround/model/PPM.py`

---

> > > > ### Author Response · Authors · 2025-11-19
> > > > **Rebuttal by Authors**
> > > >
> > > > **3. Complexity of RL-based Solution**
> > > >
> > > > Reinforcement Learning (RL) is introduced to **address the hard-boundary problem** in layer selection, enabling the model to explicitly choose a layer (e.g., layer 26) in a soft, differentiable manner during training. Because hard-boundary layer selection is non-differentiable, **using RL is necessary**.
> > > >
> > > > The Reviewer (Bxic) also has raised the same concerns through two questions:  (1) *"Why can't dynamic layer selection be trained with a simple cross-entropy loss?"* and (2) *"Why not use soft-gating without an explicit loss on the gates?"*  As suggested by the reviewer, (1) + (2) yields a simpler alternative, which we refer to as **"CE + soft-gating."**  In this design, the overlap (IoU) between each layer’s similarity map (layers 0–31) and the ground-truth mask is used to assign the layer-selection label. A layer-selection classifier is then trained, and its output scores are used as soft gates to weight and fuse the transformer layers (0–31) into a unified representation, which is subsequently fed into SAM for mask decoding.  However, **CE + soft-gating** suffers from the following issues:
> > > >
> > > > 1) **CE + soft-gating is numerically suboptimal.** Consider a scenario after training for 1–2 epochs where the classifier predicts the following scores for layers 0–31: ($[0.01, 0.004, \ldots, 0.7, 0.1, 0.1]$).  Since 0.7 is the maximum score, the 29-th layer is identified as the *best* layer.  However, if we use the full score vector ($[0.01, 0.004, \ldots, 0.7, 0.1, 0.1]$) as soft-gating weights to fuse the embeddings of layers 0–31, the best embedding from layer 29 will inevitably be mixed with features from all other layers (0–28 and 30–31), making it no longer optimal. Soft-gating inherently prevents the remaining layers from receiving zero weight.  In fact, as shown in Figure 6 of the UGround Appendix (and also discussed in [NAACL 2025][1]), transformer layers are activated only at a few specific depths. Assigning non-zero weights to inactive layers dilutes the contribution of these key activated layers.  **Therefore, a hard boundary is required—that is, explicitly selecting a single layer (e.g., assigning 100% weight to layer 29) rather than distributing fractional weights (e.g., 60% on layer 29). This is inherently a discrete decision problem, for which reinforcement learning (RL) provides a principled and trainable solution.**
> > > >
> > > > [1] From Redundancy to Relevance: Enhancing Explainability in Multimodal Large Language Models, NAACL 2025.
> > > >
> > > > 2) **The training objective of the layer-selection classifier (i.e., CE loss) does not align with the ultimate goal of optimizing the gates through backpropagation from the downstream SAM mask decoding (i.e., use soft-gating without an explicit loss on the gates)**.  The layer-selection classifier explicitly trains the gate to predict which layer is the most important, whereas using soft weights to fuse all layers aims to optimize the overall representation, without explicitly enforcing the gate to select the optimal layer. In fact, during backpropagation, this can interfere with the training of the layer-selection classifier.
> > > >
> > > > 3) **Soft-gating may ignore inter-layer dependencies in the transformer**. By weighting and summing all layers, it actually performs a soft ensemble, disrupting the original layer-by-layer structure of the transformer. In addition, it reduces interpretability.
> > > >
> > > > [2] ResNet: Deep Residual Learning for Image Recognition, CVPR 2016.
> > > >
> > > > 4) **Computational cost: CE + Soft Gating vs. RL** In fact, based on the analysis of computational costs, the RL-based solution is lightweight and does not add significant complexity, aside from requiring some extra knowledge. Given $\boldsymbol{h}\_{seg} \in \mathbb{R}^{4096}$ and $\boldsymbol{h}\_{img} \in \mathbb{R}^{576 \times 4096}$, the similarity map is computed as:  $\boldsymbol{h}\_{seg} \times \boldsymbol{h}\_{img}^{T} = (1 \times 4096) \times (576 \times 4096)^{T} = 576 $,
> > > > followed by reshaping into $24 \times 24$, and then interpolating to $256 \times 256$ to obtain $\mathcal{M}$.  If using the CE + soft-gating approach, one must compute the similarity map for **all 32 layers** of LLaVA, resulting in **32 × 256 × 256** maps, and then compute the overlap (IoU) with the ground-truth mask as the supervision label.  In contrast, with RL, we compute $\boldsymbol{h}\_{seg} \times \boldsymbol{w}^{T} = (1 \times 4096) \times (32 \times 4096)^{T} = 32$, yielding a 32-dimensional **similarity logits** vector, followed by REINFORCE to select the layer index, e.g., ($\ell^* = 28$). $\boldsymbol{w}^T \in \mathbb{R}^{32 \times 4096}$ denotes the layer-gating weights, which are the only learnable parameters in our UGround. The reward $r$ only requires computing the similarity map for **a single layer**. Here we present the core implementation of the **dynamic layer selection module (SSC)**.

---

> > > > > ### Author Response · Authors · 2025-11-19
> > > > > **Rebuttal by Authors**
> > > > >
> > > > > ```
> > > > >     def _policy_walker_mode3(self):
> > > > >
> > > > >         # Build per-token layer logits from detached features (condition x) and learnable per-layer weights
> > > > >         # seg_token_embeds: [N_seg, L, D], layer_gate_W_similarity: [L, D]
> > > > >         # logits: [N_seg, L]
> > > > >         logits_similarity = torch.einsum('nld,ld->nl', self.seg_token_embeds.detach(), self._layer_gate_W_similarity)
> > > > >         # Compute mean layer probabilities across tokens for critic input
> > > > >         if logits_similarity.size(0) > 0:
> > > > >             layer_probs = torch.softmax(logits_similarity, dim=-1)  # [N_seg, L]
> > > > >             self.last_layer_probs_mean = layer_probs.mean(dim=0).detach()  # [L]
> > > > >         else:
> > > > >             self.last_layer_probs_mean = None
> > > > >             return self._policy_walker_mode1()
> > > > >
> > > > >         # Per-token sampling from categorical(logits)
> > > > >         dist = torch.distributions.Categorical(logits=logits_similarity)
> > > > >         actions = dist.sample()  # [N_seg]
> > > > >         self.log_probs = dist.log_prob(actions)  # [N_seg]
> > > > >         batch_indices = torch.arange(self.seg_token_embeds.size(0), device=self.seg_token_embeds.device)
> > > > >         seg_token_embeds_for_similarity = self.seg_token_embeds[batch_indices, actions]  # [N_seg, D]
> > > > >         seg_image_token_embeds_for_similarity = self.seg_image_token_embeds[batch_indices, actions]  # [N_seg, P, D]
> > > > >         # Use the same chosen layer for SAM branch for consistency
> > > > >         seg_token_embeds_for_sam = seg_token_embeds_for_similarity
> > > > >         return seg_token_embeds_for_similarity, seg_image_token_embeds_for_similarity, seg_token_embeds_for_sam
> > > > > ```
> > > > > See also: `UGround/model/PPM.py`
> > > > >
> > > > > **Also, we only use the most basic reinforcement learning formulation, REINFORCE, rather than more advanced methods such as PPO, or GRPO. The only additional trainable parameters we introduce are the 32×4096 parameters (PPM.py: Lines 32–34)**.
> > > > >
> > > > > See also: `UGround/model/PPM.py`
> > > > > ```
> > > > >   31      # Layer gating weights (initialized with provided hidden_dim)
> > > > >   32      self._layer_gate_W_similarity = nn.Parameter(
> > > > >   33                 torch.ones(self.num_layers, self.hidden_dim)
> > > > >   34      )   # self.num_layers=32, self.hidden_dim=4096
> > > > > ```

---

> > > > > > ### Author Response · Authors · 2025-11-19
> > > > > > **Rebuttal by Authors**
> > > > > >
> > > > > > **Questions**
> > > > > > >1. What is the practical impact on inference-time latency and memory usage compared to the baseline model?
> > > > > >
> > > > > > **The training and inference costs of UGround are very low and can be considered negligible**.  We report UGround’s training and inference costs on a single NVIDIA A100-SXM4-40GB GPU. We train on the ReasonSeg train split over 400 samples for 1 epoch (200 steps), with a batch size of 2. *Training Latency (s)* refers to the average time per iteration during the 200-step training process (including forward, backward, and optimizer step), and *Memory Usage* refers to the MaxMemAllocated observed within the 200 steps. We also report the model’s *Trainable Percentage* (Trainable Parameters / Total Parameters). For inference, we evaluate on the ReasonSeg test split over 799 samples, with a batch size of 1. *Speed (FPS)* refers to the average frames per second computed over the 799 samples.
> > > > > >
> > > > > > **Comparing the training cost of our UGround to state-of-the-art methods**.
> > > > > > | Model   | Batch Size | Training Latency (s) | Memory Usage (GB) | Trainable Percentage | Trainable Parameters | Total Parameters |
> > > > > > | :-------  | :------------: | :------------: | :----------:  | :--------------------: | :--------------------: | :----------------: |
> > > > > > | LISA [3]    | 2 |1.26         | 23.68      | 3.74%                | 288.25M (288,251,364)          | 7.71B (7,708,933,424)    |
> > > > > > | SESAME [2] | 2 | 1.11         | 23.15      | 3.73%                | 288.25M (288,251,364)          | 7.73 B (7,725,714,736)    |
> > > > > > | GSVA    | 2         | 2.13      |   25.73        | 3.73%                | 288.26M (288,259,556)          | 7.73B (7,726,050,608)    |
> > > > > > | PixelLM | 2         | 1.22      |   23.02       | 5.25%                | 375.72M (375,724,772)          | 7.16B (7,157,608,240)    |
> > > > > > | UGround | 2 | 2.97         | 28.16      | 3.72%                | 288.40M (288,400,944)          | 7.75 B (7,747,165,488)    |
> > > > > >
> > > > > > **Comparing the runtime speed of our UGround to state-of-the-art methods**.
> > > > > > | Model        | GSVA [1] | SESAME [2] | LISA [3] |PixelLM [4]|UGround|
> > > > > > |:------------:|:--------:|:----------:|:-----------:|:-----------:|:-----------:|
> > > > > > | Speed (FPS)  |   3.98   |    4.64    |   4.68      |    9.24     |    4.12     |
> > > > > >
> > > > > > [1] GSVA: Generalized Segmentation via Multimodal Large Language Models, CVPR 2024.
> > > > > >
> > > > > > [2] SESAME: See, Say, and Segment: Teaching LMMs to Overcome False Premises, CVPR 2024.
> > > > > >
> > > > > > [3] LISA: Reasoning Segmentation via Large Language Model, CVPR 2024.
> > > > > >
> > > > > > [4] PixelLM: Pixel Reasoning with Large Multimodal Model, CVPR 2024.

---

> > > > > > > ### Author Response · Authors · 2025-11-19
> > > > > > > **Rebuttal by Authors**
> > > > > > >
> > > > > > > >2. The layer selection policy is conditioned solely on the textual <SEG> token's representations. Have the authors considered or experimented with also conditioning the policy on visual information (e.g., a pooled representation of image tokens)?
> > > > > > >
> > > > > > > We agree.  Currently, the layer-selection policy $\pi_\theta(\ell \mid \mathcal{H}_{t^{*}})$ is conditioned only on the hidden states of the [SEG] token. In a future journal extension, we plan to advance UGround by conditioning the policy on image token representations as well. It is **important to note that image token representations can serve only as *auxiliary* information (Indeed, the image itself could plausibly provide more informative cues for layer selection.)—they cannot condition the policy on their own**. In a multi-target setting, where a single sample contains multiple [SEG] tokens, we need to select a dedicated layer for each [SEG] token. In this case, the image token representations act as shared information across targets: they cannot independently determine the layer for each [SEG] token, but they may help provide a more robust and adaptive selection mechanism when used as supplementary features.  In our implementation, incorporating image token representations into the policy conditioning is straightforward (**Line 140**).
> > > > > > > ```
> > > > > > > 135    def _policy_walker_mode3(self):
> > > > > > > 136
> > > > > > > 137       # Build per-token layer logits from detached features (condition x) and learnable per-layer weights
> > > > > > > 138       # seg_token_embeds: [N_seg, L, D], layer_gate_W_similarity: [L, D]
> > > > > > > 139       # logits: [N_seg, L]
> > > > > > > 140       logits_similarity = torch.einsum('nld,ld->nl', self.seg_token_embeds.detach(), self._layer_gate_W_similarity)
> > > > > > > 141       # Compute mean layer probabilities across tokens for critic input
> > > > > > > 142       if logits_similarity.size(0) > 0:
> > > > > > > 143               layer_probs = torch.softmax(logits_similarity, dim=-1)  # [N_seg, L]
> > > > > > > 144               self.last_layer_probs_mean = layer_probs.mean(dim=0).detach()  # [L]
> > > > > > > 145       else:
> > > > > > > 146              self.last_layer_probs_mean = None
> > > > > > > 147              return self._policy_walker_mode1()
> > > > > > > 148
> > > > > > > 149       # Per-token sampling from categorical(logits)
> > > > > > > 150       dist = torch.distributions.Categorical(logits=logits_similarity)
> > > > > > > 151       actions = dist.sample()  # [N_seg]
> > > > > > > 152       self.log_probs = dist.log_prob(actions)  # [N_seg]
> > > > > > > 153       batch_indices = torch.arange(self.seg_token_embeds.size(0), device=self.seg_token_embeds.device)
> > > > > > > 154       seg_token_embeds_for_similarity = self.seg_token_embeds[batch_indices, actions]  # [N_seg, D]
> > > > > > > 155       seg_image_token_embeds_for_similarity = self.seg_image_token_embeds[batch_indices, actions]  # [N_seg, P, D]
> > > > > > > 156       # Use the same chosen layer for SAM branch for consistency
> > > > > > > 157       seg_token_embeds_for_sam = seg_token_embeds_for_similarity
> > > > > > > 158       return seg_token_embeds_for_similarity, seg_image_token_embeds_for_similarity, seg_token_embeds_for_sam
> > > > > > > ```
> > > > > > > See also: `UGround/model/PPM.py`
> > > > > > >
> > > > > > > >3. The reward `r` is derived from the similarity map's alignment with the ground-truth mask. Have the authors explored alternative reward signals, such as the final segmentation IoU score generated by SAM? Such a reward might more directly optimize for the final task performance.
> > > > > > >
> > > > > > > We agree. In our UGround, the data flow pipeline is
> > > > > > >
> > > > > > > input: ($x\_{img}$, $x\_{txt}$) $\rightarrow$ LLaVA (layers: $\ell=0,1,..., 31$) $\rightarrow$ **(a) layer select**: e.g., $\ell^*$=18 $\rightarrow$ **(b) MasP**:$\mathcal{M}^{(18)}$, $\boldsymbol{h}\_{seg}^{\left( 18 \right)}\in \mathbb{R} ^{4096}$ $\rightarrow$ SAM: $\hat{\mathbf{M}}$
> > > > > > >
> > > > > > > We leverage similarity map $\mathcal{M}^{(18)}$ in three ways: as a logit mask to prompt SAM (prompt) for mask generation $\hat{\mathbf{M}}$, as a loss constraint to guide the model on where to "attend" (constraint), and as a policy reward $r$ (derived from the similarity map's alignment with the ground-truth mask) for dynamic layer selection (signal).
> > > > > > >
> > > > > > > Indeed, using the final segmentation IoU score generated by SAM as a reward $r$ might more directly optimize for task performance. This is valuable, and we will further validate it and report the corresponding results in an upcoming version if it works. Note that LLaVA and SAM are originally two separately trained models. They seek representation alignment via [SEG] as a prompt (LISA [1]) or mask as a prompt (ours), but an alignment gap does exist between them. **Experiments show that sometimes LLaVA correctly understands the query (i.e., interprets the instruction correctly), but SAM fails in segmentation (i.e., fails to execute the instruction). In such cases, the final segmentation IoU score generated by SAM may not accurately reflect the optimal layer of LLaVA**.
> > > > > > >
> > > > > > > [1] LISA: Reasoning Segmentation via Large Language Model, CVPR 2024.

---

> > > > > > > > ### Author Response · Authors · 2025-11-19
> > > > > > > > **Official Comment by Authors**
> > > > > > > >
> > > > > > > > Dear reviewer, we would like to thank you for your time and efforts. We hope that your concerns are addressed with our rebuttal. Please let us know if there are any further questions that need clarification.

---

> ### Author Response · Authors · 2025-11-26
> **The revised paper is now available.**
>
> Hi Reviewer L9cY, the revised paper is now available at [https://openreview.net/pdf?id=pWi3tvhhmx](https://openreview.net/pdf?id=pWi3tvhhmx). All modifications are clearly highlighted in **blue** throughout the manuscript.
>
> * **Complexity of RL-based Solution**: Justification for Reinforcement Learning (RL) is supported by comparative experiments (Lines 480–485) and further discussed in Lines 726–764.
>
> * **Inference Overhead and Ambiguity**: Computational costs are reported in Lines 514–524.
>
> The main paper has been carefully checked and polished.

---

### Official Review · Reviewer_aGp8 · 2025-11-01

**Soundness:** 2
**Presentation:** 1
**Contribution:** 1
**Rating:** 2
**Confidence:** 4

**Summary:**

This paper addresses the task of Visual Grounding of Referring Expressions by proposing a transformer-based architecture. UGround learns similarity maps between image tokens and the <SEG> token, which are then used as prompts for SAM. The proposed method learns a policy to determine which intermediate transformer layer to use, rather than fixing the output of the last hidden layer. This paper achieves competitive results in ReasonSeg, RefCOCO/+/g, and GRES datasets and reports ablation experiments.

**Strengths:**

This review evaluates the paper's quality based on the following criteria: task relevance, related work, technical novelty, technical correctness, experimental validation, writing and presentation, and reproducibility. Each aspect is discussed and highlighted as a strength or a weakness in the sections below.
-    **Relevance of the task:** Visual Grounding of Referring Expressions is a highly relevant problem for the ICLR community. This paper presents competitive results on benchmark datasets for this task.
-    **Reproducibility:** The source code is included in the supplementary material.

**Weaknesses:**

-	**Writing and Presentation:** This paper misses clarity and consistency. The names of the proposed submodules are too complex to describe the actual methodological contributions of this work. Moreover, the mathematical nomenclature used in the paper is not consistent. Specifically, to which refer different variants for using similarity as mask (Line 219 – 234) is not clear.
-	**Related Work:** The Related Work section does not adequately contextualize the contributions. Specifically, it remains unclear which limitations of prior works, such as the “alignment-centric models” (Line 131), this method addresses.
-	**Experimental Validation and Technical Novelty:**
    -    This paper claims to propose a unified approach for Visual Grounding of Referring Expressions; however, it does not make clear which methodological modifications from previous works are required to fulfill this proposal.
    -    The empirical results are only marginally better or just competitive than the current state-of-the-art for this task on RefCOCO/+/g.
    -    Moreover, it is not clear how significant modifications from previous methods, like including SAM as the segmentation decoder, are influencing the experimental validation rather than the actual technical contributions of this work.

**Questions:**

1.	What specific changes or improvements make this method a "unified" approach compared to previous visual grounding methods?
2.	Which limitations of previous Large Multimodal Models for visual grounding are being addressed, and how does this method solve them?
3.	The empirical results are close to existing state-of-the-art methods. Why these results are significant enough to justify the architectural modifications?
4.	How much of the performance improvement comes from using SAM, and how much comes from the new method itself?
5.	Please make clarifications for the mathematical nomenclature.

---

> ### Author Response · Authors · 2025-11-19
> **Rebuttal by Authors**
>
> **1. Writing and Presentation**
> >This paper misses clarity and consistency. The names of the proposed submodules are too complex to describe the actual methodological contributions of this work. Moreover, the mathematical nomenclature used in the paper is not consistent. Specifically, to which refer different variants for using similarity as mask (Line 219 – 234) is not clear.
>
> **The *names* of the proposed submodules are not complex**: We introduce two submodules—(1) **Stochastic Skip Connection (SSC)** and (2) **Mask as Prompt (MasP)**. The term *"Skip Connection"* is borrowed from ResNet [1]; since we incorporate RL-based stochastic sampling, the "skip" operation becomes random, hence the name **Stochastic Skip Connection (SSC)**. The term **Mask as Prompt** is defined in contrast to *[SEG] as Prompt*, preserving a symmetric naming scheme.
>
> [1] ResNet: Deep Residual Learning for Image Recognition, CVPR 2016.
>
> ---
>
> In original work LISA [1],
>
> input: ($x\_{img}$, $x\_{txt}$) $\rightarrow$ LLaVA (layers: $\ell=0,1,..., 31$) $\rightarrow$ **(c)fixed last layer:** $\boldsymbol{h}\_{seg}^{\left( 31 \right)}\in \mathbb{R} ^{4096}$ $\rightarrow$ SAM: $\hat{\mathbf{M}}$
>
> In our UGround, we advance LISA [1] via components **(a): Stochastic Skip Connection (SSC)** and **(b): Mask
> as Prompt (MasP)**:
>
> input: ($x\_{img}$, $x\_{txt}$) $\rightarrow$ LLaVA (layers: $\ell=0,1,..., 31$) $\rightarrow$ **(a) layer select**: e.g., $\ell^*$=18 $\rightarrow$ **(b) MasP**:$\mathcal{M}^{(18)}$, $\boldsymbol{h}\_{seg}^{\left( 18 \right)}\in \mathbb{R} ^{4096}$ $\rightarrow$ SAM: $\hat{\mathbf{M}}$
>
> **The SSC and MasP modules are easy to understand, and the core method section (Sec. 4.1 Policy-Prompted Masking: Lines 256–350) spans only 1 page**.  In fact, we only use the most basic reinforcement learning formulation, REINFORCE, rather than more advanced methods such as PPO, or GRPO.
>
> **The mathematical notation in this paper is consistent and rigorous**. "to which refer different variants for using similarity as mask (Line 219 – 234) is not clear",  $\mathcal{S} _{\mathrm{IoU}}$ is a typo, which can be inferred from the table caption itself, and it does not affect the readers’ understanding of the table. $\mathcal{S}\_{\mathrm{IoU}}\rightarrow \mathcal{M}\_{\mathrm{IoU}}$.
>
> We would like to provide further clarification on Lines 219–234 to facilitate reviewer's understanding:
>
> "An adapted SAM" — here, "adapted" means that the SAM model can receive text (i.e., [seg] token) as a prompt. In the original SAM paper, it does not accept text as a prompt; LISA [1] modifies it. LISA's successors, such as SESAME [2], GSVA [3], and PixelLM [4], all adopt this paradigm.  **The Analysis in Sec. 3.2 (Lines 219-234: "Why Similarity as Mask.") serves to explain the motivation, thereby leading to the introduction of our method.**  LISA and its successors, such as SESAME, GSVA, and PixelLM, all use [seg] as a prompt, which has been demonstrated to be feasible. In contrast, we aim to use the logit mask (similarity map) as a prompt, and **we want to investigate whether similarity map is feasible**. To this end, we implemented the following: to measure the consistency between the similarity map and the ground truth mask, we compute the $\mathcal{M}\_{\mathrm{IoU}}$ between them; to probe whether SAM can understand the semantics of the similarity map, we directly use the similarity map to prompt the original SAM (without training), denoted as $\mathcal{M}\_{\mathrm{prompt}}$.
>
> [1] LISA: Reasoning Segmentation via Large Language Model, CVPR 2024.
>
> [2] SESAME: See, Say, and Segment: Teaching LMMs to Overcome False Premises, CVPR 2024.
>
> [3] GSVA: Generalized Segmentation via Multimodal Large Language Models, CVPR 2024.
>
> [4] PixelLM: Pixel Reasoning with Large Multimodal Model, CVPR 2024.
>
>
> **We want to claim our core contribution**: 1) **Stochastic Skip Connection(SSC)** and 2) **Mask as Prompt (MasP)**, as stated in our paper (**Abstract, Lines 18-26; Sec. 1 introduction, Lines 95-107**). SSC is a reinforcement learning policy that, via stochastic sampling, allows each [SEG] token to slide across unrolled transformer layers, enabling dynamic layer selection at which it connects to the vision model (e.g., SAM) **in a skip-connection fashion ("skip-connection-like'')**[1]. By skipping subsequent ones of $\ell^*$, we can ''cheat'' this telephone game (Main text: Lines 82-94) by letting the final participant (SAM) tap into intermediate layers (e.g., 1–39) in advance. Across several forward pass, **such mechanism virtually connects all layers to SAM while activating only one pass at a time ("dropout-like'')**. Given the selected hidden layer, MasP uses the similarity map derived from the [SEG] token and image tokens as a soft logit mask to prompt SAM for mask generation, **offering explicit spatial cues through its activation regions**.
>
> [1] ResNet: Deep Residual Learning for Image Recognition, CVPR 2016.

---

> ### Author Response · Authors · 2025-11-19
> **Rebuttal by Authors**
>
> **2. Related Work**
> >The Related Work section does not adequately contextualize the contributions. Specifically, it remains unclear which limitations of prior works, such as the “alignment-centric models” (Line 131), this method addresses.
>
> Regarding the concern *“unclear which limitations of prior works”*, we view the “limitations of prior works” at two levels: **method-level** and **architecture-level**.
>
> **At the method-level**, we have already explicitly pointed out the limitations in the abstract (Lines 14–19):  *“UGround addresses two primary challenges posed by the prevailing paradigm: (1) its reliance on the fixed last hidden layer, which sequentially amplifies cumulative errors arising from layer-by-layer propagation without intermediate correction, and (2) its use of <SEG> as a prompt, which implicitly projects textual embeddings into visual space without explicit spatial cues (e.g., coordinates).”* In the Introduction (Lines 74–94), we further elaborate on these two points. In Sec. 2 Related Work, 1) Large Multimodal Models.  Since our method relies on Large Multimodal Models for instruction text understanding, we briefly review the progress of existing Large Multimodal Models and categorize them into *fusion-centric models* and *alignment-centric models*.  As stated in our paper (Sec. 2 Related Work, Lines 131-135), *"Alignment-centric models project the outputs of frozen or lightly fine-tuned unimodal encoders (e.g., ViT for vision, LLaMA/GPT for text) into a shared latent space via a lightweight alignment adapter (e.g., projectors, Q-Formers),"* among which LLaVA is a representative example.  **As stated in our paper (Sec. 2 Related Work, Line 141)**: *"our UGround builds upon LLaVA (Liu et al., 2024a) for visual autoregressive modeling."*
>
> **At the architecture-level**, we clearly point out the problems that exist in current unified frameworks. In Sec. 2 Related Work, 2) **Unified Segmentation Models.** Depending on the semantic granularity of joint tasks, we divide the literature into two groups: *versatility-oriented segmentation* and *attribute-oriented segmentation*. We, for the first time, unify visual grounding within a single framework from an attribute perspective.  **As stated in our paper (Sec. 2 Related Work, Lines 153–161)**:
> *"Closest to our work, PixelLM (Ren et al., 2024) cannot tackle empty targets, while GSVA (Xia et al., 2024) fails to handle multi-target reasoning. Attribute-oriented unification is more fundamental, as it can be applied from a versatile perspective to any specialized task. Yet, we observe that little research has looked into unifying existing tasks within a single framework from an attribute perspective, we therefore present UGround."*

---

> ### Author Response · Authors · 2025-11-19
> **Rebuttal by Authors**
>
> **3. Experimental Validation and Technical Novelty**
>
> >1. This paper claims to propose a unified approach for Visual Grounding of Referring Expressions; however, it does not make clear which methodological modifications from previous works are required to fulfill this proposal.
>
> Our core contribution are twofold: **1) architecture-level** and **2) method-level**.
>
> **As stated in our paper (Sec. 2 Intruduction, Lines 108–115)**:
>
> • We have "unified" visual grounding within a single framework from an attribute perspective. Previously, existing works either focus on explicit rather than implicit expressions, or handle single-target instead of multi-target scenarios, or respond solely to positive queries without rejecting false premises (empty target).
>
> • We have "unrolled" stacked transformers, letting vision decoder tap into intermediate layers in a stochastic skip-connection fashion, in the way of "mask as prompt". Importantly, we explicitly supervise the prompted mask (similarity map) against the ground-truth mask to further guide the model on where to attend.
>
> **At the architecture-level**, we unify multiple complex visual grounding tasks within a single, coherent framework. This involves issues at the implementation level, such as designing task-specific dataloaders for different datasets, supporting multiple [SEG] tokens for multi-target scenarios, and handling false premises by adding support for empty targets through [REJ] token. **These details can be found in the implementation code provided in the [supplementary material](https://openreview.net/attachment?id=pWi3tvhhmx&name=supplementary_material), and the directory structure of the code is shown in the appendix (Lines 657–681)**.
>
> However, unifying multiple complex visual grounding tasks within a single, coherent architecture alone is insufficient; we also need to addresses two primary challenges posed by the prevailing paradigm (**In the abstract, Lines 14-19**): "(1) its reliance on the fixed last hidden layer, which sequentially amplifies cumulative errors arising from layer-by-layer propagation without intermediate correction, and (2) its use of <SEG> as a prompt, which implicitly projects textual embeddings into visual space without explicit spatial cues (e.g., coordinates)". As stated in our paper (Sec. 2 Introduction, Lines 47–54): "Can we design a unified architecture to bridge this gap?" This naturally leads to the method-level contributions below.
>
> **At the method-level**,  Central to UGround is two key components: 1) **Stochastic Skip Connection(SSC)** and 2) **Mask as Prompt(MasP)**. SSC is a reinforcement learning policy that, via stochastic sampling, allows each [SEG] token to slide across unrolled transformer layers, enabling dynamic layer selection at which it connects to the vision model (e.g., SAM) **in a skip-connection fashion ("skip-connection-like'')**[2]. By skip subsequent ones of $\ell^*$, we can ''cheat'' this telephone game (Main text: Lines 82-94) by letting the final participant (SAM) tap into intermediate layers (e.g., 1–39) in advance. Across several forward pass, **such mechanism virtually connects all layers to SAM while activating only one pass at a time ("dropout-like'')**. Given the selected hidden layer, MasP uses the similarity map derived from the [SEG] token and image tokens as a soft logit mask to prompt SAM for mask generation, **offering explicit spatial cues through its activation regions**.
>
> For simplicity, we provide the UGround data flow pipeline below.
>
> In original work LISA [1],
>
> input: ($x\_{img}$, $x\_{txt}$) $\rightarrow$ LLaVA (layers: $\ell=0,1,..., 31$) $\rightarrow$ **(c)fixed last layer:** $\boldsymbol{h}\_{seg}^{\left( 31 \right)}\in \mathbb{R} ^{4096}$ $\rightarrow$ SAM: $\hat{\mathbf{M}}$
>
> In our UGround, we advance LISA [1] via components **(a): Stochastic Skip Connection (SSC)** and **(b): Mask
> as Prompt (MasP)**:
>
> input: ($x\_{img}$, $x\_{txt}$) $\rightarrow$ LLaVA (layers: $\ell=0,1,..., 31$) $\rightarrow$ **(a) layer select**: e.g., $\ell^*$=18 $\rightarrow$ **(b) MasP**:$\mathcal{M}^{(18)}$, $\boldsymbol{h}\_{seg}^{\left( 18 \right)}\in \mathbb{R} ^{4096}$ $\rightarrow$ SAM: $\hat{\mathbf{M}}$
>
> [1] LISA: Reasoning Segmentation via Large Language Model, CVPR 2024.

---

> > ### Author Response · Authors · 2025-11-19
> > **Rebuttal by Authors**
> >
> > >2. The empirical results are only marginally better or just competitive than the current state-of-the-art for this task on RefCOCO/+/g.
> >
> > We report results on three datasets: ReasonSeg, RefCOCO, and gRefCOCO. **ReasonSeg serves as our main reported dataset**, showing a significant improvement (ReasonSeg test: gIoU $60.3 \rightarrow 63.6$, cIoU $60.0 \rightarrow 65.4$). **RefCOCO and gRefCOCO are used for generalization experiments**. RefCOCO, as a classical dataset, has already been extensively fine-tuned, so the fact that UGround achieves **“marginally better or just competitive performance compared to the current state-of-the-art for this task on RefCOCO/+/g”** is non-trivial. In addition, we use significantly less training data for RefCOCO (~10k images) compared to prior state-of-the-art. Finally, we also observe a notable improvement on the multi-target dataset gRefCOCO (gRefCOCO val: gIoU $66.47 \rightarrow 72.46$, cIoU $63.29 \rightarrow 65.56$, N-acc. $62.43 \rightarrow 74.53$).
> >
> > Table 3. **Benchmarking reasoning segmentation models on the ReasonSeg dataset**, sorted in ascending order of cIoU on the test set. *: reproduced from official models. ft: fine-tuned on 239 samples.
> >
> > | Method | val gIoU | val cIoU | test short gIoU | test short cIoU | test long gIoU | test long cIoU | test overall gIoU | test overall cIoU |
> > |------|:--------:|:--------:|:---------------:|:---------------:|:---------------:|:---------------:|:----------------:|:----------------:|
> > | X-Decoder | 22.6 | 17.9 | 20.4 | 11.6 | 22.2 | 17.5 | 21.7 | 16.3 |
> > | Grounded-SAM | 26.0 | 14.5 | 17.8 | 10.8 | 22.4 | 18.6 | 21.3 | 16.4 |
> > | SEEM | 25.5 | 21.2 | 20.1 | 11.5 | 25.6 | 20.8 | 24.3 | 18.7 |
> > | OVSeg | 28.5 | 18.6 | 18.0 | 15.5 | 28.7 | 22.5 | 26.1 | 20.8 |
> > | GRES | 22.4 | 19.9 | 17.6 | 15.0 | 22.6 | 23.8 | 21.3 | 22.0 |
> > | **7B Models** |  |  |  |  |  |  |  |  |
> > | *SESAME | 40.3 | 41.6 | 28.9 | 26.3 | 37.3 | 31.9 | 34.9 | 30.7 |
> > | LLaVA1.5-7B + OVSeg | 38.2 | 23.5 | 24.2 | 18.7 | 44.6 | 37.1 | 39.7 | 31.8 |
> > | *GSVA | 45.6 | 41.5 | 37.9 | 36.5 | 44.3 | 46.0 | 42.8 | 43.8 |
> > | *PixelLM | 49.7 | 49.6 | 39.5 | 38.8 | 49.5 | 45.6 | 47.1 | 44.3 |
> > | LISA-7B | 52.9 | 54.0 | 40.6 | 40.6 | 49.4 | 51.0 | 47.3 | 48.4 |
> > | HyperSeg-3B | 59.2 | 56.7 | - | - | - | - | - | - |
> > | VISA-7B | 52.7 | 57.8 | - | - | - | - | - | - |
> > | VideoLISA-3.8B | 61.4 | 67.1 | 43.8 | 42.7 | 56.9 | 57.7 | 53.8 | 54.4 |
> > | LISA-7B-LLaVA1.5 (ft) | 61.3 | 62.9 | 48.3 | 46.3 | 57.9 | 59.7 | 55.6 | 56.9 |
> > | READ-7B-LLaVA1.5 (ft) | 59.8 | 67.6 | 52.6 | 49.5 | 60.4 | 61.0 | 58.5 | 58.6 |
> > | LISA++-7B-LLaVA1.5 (ft) | 64.2 | 68.1 | 49.6 | **51.1** | 59.3 | 61.7 | 57.0 | 59.5 |
> > | RSVP-GPT | 64.7 | 63.1 | **55.4** | 50.4 | 61.9 | 62.5 | 60.3 | 60.0 |
> > | UGround-7B-LLaVA1.5 (ft) | **66.1** | **72.1** | 55.1 | 48.5 | **66.3** | **70.2** | **63.6** | **65.4** |
> > | **13B Models** |  |  |  |  |  |  |  |  |
> > | LLaVA1.5-13B + OVSeg | 37.9 | 26.4 | 27.1 | 19.4 | 46.1 | 40.6 | 41.5 | 34.1 |
> > | LISA-13B-LLaVA1.5 | 57.7 | 60.3 | 50.8 | 50.0 | 54.7 | 50.9 | 53.8 | 50.8 |
> > | LISA-13B-LLaVA1.5 (ft) | 65.0 | 72.9 | 55.4 | 50.6 | 63.2 | 65.3 | 61.3 | 62.2 |
> > | READ-13B-LLaVA1.5 (ft) | - | - | 55.4 | **53.7** | 64.4 | 65.1 | 62.2 | 62.8 |
> > | UGround-13B-LLaVA1.5 (ft) | **67.9** | **74.9** | **57.2** | 50.9 | **67.5** | **69.4** | **65.0** | **65.5** |

---

> > > ### Author Response · Authors · 2025-11-19
> > > **Rebuttal by Authors**
> > >
> > > Table 4. **Benchmarking referring segmentation models on the RefCOCO(+/g) dataset**,  sorted in ascending order of cIoU on the RefCOCOg val set.
> > >
> > > | Method | refCOCO val | refCOCO testA | refCOCO testB | refCOCO+ val | refCOCO+ testA | refCOCO+ testB | refCOCOg val(U) | refCOCOg test(U) |
> > > |------|:-----------:|:-------------:|:-------------:|:------------:|:--------------:|:--------------:|:---------------:|:----------------:|
> > > | MCN | 62.4 | 64.2 | 59.7 | 50.6 | 55.0 | 44.7 | 49.2 | 49.4 |
> > > | VLT | 67.5 | 70.5 | 65.2 | 56.3 | 61.0 | 50.1 | 55.0 | 57.7 |
> > > | CRIS | 70.5 | 73.2 | 66.1 | 62.3 | 68.1 | 53.7 | 59.9 | 60.4 |
> > > | LAVT | 72.7 | 75.8 | 68.8 | 62.1 | 68.4 | 55.1 | 61.2 | 62.1 |
> > > | X-Decoder | - | - | - | - | - | - | 64.6 | - |
> > > | ReLA | 73.8 | 76.5 | 70.2 | 66.0 | 71.0 | 57.7 | 65.0 | 66.0 |
> > > | SEEM | - | - | - | - | - | - | 65.7 | - |
> > > | Segment Anyword | 55.3 | 47.9 | 66.0 | 55.6 | 47.4 | 67.0 | 58.4 | 60.1 |
> > > | VISA-7B | 72.4 | 75.5 | 68.1 | 59.8 | 64.8 | 53.1 | 65.5 | 66.4 |
> > > | SESAME | 74.7 | - | - | 64.9 | - | - | 66.1 | - |
> > > | LISA-7B | 74.9 | 79.1 | 72.3 | 65.1 | 70.8 | 58.1 | 67.9 | 70.6 |
> > > | PixelLM-7B | 73.0 | 76.5 | 68.2 | 66.3 | 71.7 | 58.3 | 69.3 | 70.5 |
> > > | READ-7B | 78.1 | 80.2 | 73.2 | 68.4 | 73.7 | 60.4 | 70.1 | 71.4 |
> > > | SegLLM-7B | 80.2 | 81.5 | 75.4 | 70.3 | 73.0 | 62.5 | 72.6 | 73.6 |
> > > | GSVA-7B | 77.2 | 78.9 | 73.5 | 65.9 | 69.6 | 59.8 | 72.7 | 73.3 |
> > > | OMG-LLaVA | 78.0 | 80.3 | 74.1 | 69.1 | 73.1 | 63.0 | 72.9 | 72.9 |
> > > | GLaMM-7B | 79.5 | 83.2 | 76.9 | 72.6 | **78.7** | 64.6 | 74.2 | 74.9 |
> > > | POPEN-7B | 79.3 | 82.0 | 74.1 | **73.1** | 77.0 | 65.1 | **75.4** | 75.6 |
> > > | UGround-7B | **80.6** | **83.5** | **77.7** | 72.8 | 77.5 | **65.6** | 74.7 | **76.1** |
> > >
> > > Table 5. **Benchmarking generalized referring expression segmentation (GRES) models on the gRefCOCO** dataset, sorted in ascending order of cIoU on the gRefCOCO validation set. Values are taken from~\citep{liu2023gres}.  N-acc.: the accuracy of correct null-target classification. ft: fine-tuned on gRefCOCO training set.
> > >
> > > | Method | Validation gIoU | Validation cIoU | Validation N-acc. | Test A gIoU | Test A cIoU | Test A N-acc. | Test B gIoU | Test B cIoU | Test B N-acc. |
> > > |------|:---------------:|:---------------:|:----------------:|:-----------:|:-----------:|:--------------:|:-----------:|:-----------:|:--------------:|
> > > | MattNet | 48.24 | 47.51 | 41.15 | 59.30 | 58.66 | 44.04 | 46.14 | 45.33 | 41.32 |
> > > | LTS | 52.70 | 52.30 | - | 62.64 | 61.87 | - | 50.42 | 49.96 | - |
> > > | VLT | 52.00 | 52.51 | 47.17 | 63.20 | 62.19 | 48.74 | 50.88 | 50.52 | 47.82 |
> > > | CRIS | 56.27 | 55.34 | - | 63.42 | 63.82 | - | 51.79 | 51.04 | - |
> > > | LAVT | 58.40 | 57.64 | 49.32 | 65.90 | 65.32 | 49.25 | 55.83 | 55.04 | 48.46 |
> > > | ReLA | 63.60 | 62.42 | 56.37 | 70.03 | 69.26 | 59.02 | 61.02 | 59.88 | 58.40 |
> > > | **7B Models** | | | | | | | | | |
> > > | LISA-Vicuna-7B | 32.21 | 38.72 | 2.71 | 48.54 | 52.55 | 6.37 | 39.65 | 44.79 | 5.00 |
> > > | GSVA-Vicuna-7B | 63.32 | 61.70 | 56.45 | 70.11 | 69.23 | 63.50 | 61.34 | 60.26 | 58.42 |
> > > | LISA-Vicuna-7B (ft) | 61.63 | 61.76 | 54.67 | 66.27 | 68.50 | 50.01 | 58.84 | 60.63 | 51.91 |
> > > | GSVA-Vicuna-7B (ft) | 66.47 | 63.29 | 62.43 | 71.08 | 69.93 | 65.31 | 62.23 | 60.47 | 60.56 |
> > > | UGround-LLaVA1.5-7B (ft) | **72.46** | **65.56** | **74.53** | **74.29** | **70.87** | **73.93** | **66.85** | **61.84** | **71.22** |

---

> > > > ### Author Response · Authors · 2025-11-19
> > > > **Rebuttal by Authors**
> > > >
> > > > >Moreover, it is not clear how significant modifications from previous methods, like including SAM as the segmentation decoder, are influencing the experimental validation rather than the actual technical contributions of this work.
> > > >
> > > > **"using SAM" is not our modifications**. Such a paradigm was originally proposed by LISA [1] and has been adopted by its successors, e.g., SESAME [2] and GSVA [3]. SAM is used for mask generation/decoding, and the new method itself cannot generate masks without a decoder. Reviewer can **refer to Table 3, Line 392 and Line 394 for comparison**: LISA-7B-LLaVA1.5 (ft) vs. UGround-7B-LLaVA1.5 (ft) (ReasonSeg test: gIoU $55.6 \rightarrow 63.6$, cIoU $56.9 \rightarrow 65.4$). **Our code is built upon LISA [1], and both models use SAM as the segmentation decoder**. Here, we provide the data flow pipeline between LISA and UGround. We also recommend reviewer refer to LISA [1] for more details.
> > > >
> > > > In original work LISA [1],
> > > >
> > > > input: ($x\_{img}$, $x\_{txt}$) $\rightarrow$ LLaVA (layers: $\ell=0,1,..., 31$) $\rightarrow$ **(c)fixed last layer:** $\boldsymbol{h}\_{seg}^{\left( 31 \right)}\in \mathbb{R} ^{4096}$ $\rightarrow$ SAM: $\hat{\mathbf{M}}$
> > > >
> > > > In our UGround, we advance LISA [1] via components **(a): Stochastic Skip Connection (SSC)** and **(b): Mask
> > > > as Prompt (MasP)**:
> > > >
> > > > input: ($x\_{img}$, $x\_{txt}$) $\rightarrow$ LLaVA (layers: $\ell=0,1,..., 31$) $\rightarrow$ **(a) layer select**: e.g., $\ell^*$=18 $\rightarrow$ **(b) MasP**:$\mathcal{M}^{(18)}$, $\boldsymbol{h}\_{seg}^{\left( 18 \right)}\in \mathbb{R} ^{4096}$ $\rightarrow$ SAM: $\hat{\mathbf{M}}$
> > > >
> > > > [1] LISA: Reasoning Segmentation via Large Language Model, CVPR 2024.

---

> ### Author Response · Authors · 2025-11-19
> **Rebuttal by Authors**
>
> **Questions**
>
> >1. What specific changes or improvements make this method a "unified" approach compared to previous visual grounding methods?
>
> Our changes or improvements are twofold: **1) architecture-level** and **2) method-level**.
>
> **As stated in our paper (Sec. 2 Intruduction, Lines 108–115)**:
>
> • We have "unified" visual grounding within a single framework from an attribute perspective. Previously, existing works either focus on explicit rather than implicit expressions, or handle single-target instead of multi-target scenarios, or respond solely to positive queries without rejecting false premises (empty target).
>
> • We have "unrolled" stacked transformers, letting vision decoder tap into intermediate layers in a stochastic skip-connection fashion, in the way of "mask as prompt". Importantly, we explicitly supervise the prompted mask (similarity map) against the ground-truth mask to further guide the model on where to attend.
>
> **At the architecture-level**, we unify multiple complex visual grounding tasks within a single, coherent framework. This involves issues at the implementation level, such as designing task-specific dataloaders for different datasets, supporting multiple [SEG] tokens for multi-target scenarios, and handling false premises by adding support for empty targets through [REJ] token. **These details can be found in the implementation code provided in the [supplementary material](https://openreview.net/attachment?id=pWi3tvhhmx&name=supplementary_material), and the directory structure of the code is shown in the appendix (Lines 657–681)**.
>
> However, unifying multiple complex visual grounding tasks within a single, coherent architecture alone is insufficient; we also need to addresses two primary challenges posed by the prevailing paradigm (**In the abstract, Lines 14-19**): "(1) its reliance on the fixed last hidden layer, which sequentially amplifies cumulative errors arising from layer-by-layer propagation without intermediate correction, and (2) its use of [SEG] as a prompt, which implicitly projects textual embeddings into visual space without explicit spatial cues (e.g., coordinates)". As stated in our paper (Sec. 2 Introduction, Lines 47–54): "Can we design a unified architecture to bridge this gap?" This naturally leads to the method-level contributions below.
>
> **At the method-level**,  Central to UGround is two key components: 1) **Stochastic Skip Connection(SSC)** and 2) **Mask as Prompt(MasP)**. SSC is a reinforcement learning policy that, via stochastic sampling, allows each [SEG] token to slide across unrolled transformer layers, enabling dynamic layer selection at which it connects to the vision model (e.g., SAM) **in a skip-connection fashion ("skip-connection-like'')**[2]. By skip subsequent ones of $\ell^*$, we can ''cheat'' this telephone game (Main text: Lines 82-94) by letting the final participant (SAM) tap into intermediate layers (e.g., 1–39) in advance. Across several forward pass, **such mechanism virtually connects all layers to SAM while activating only one pass at a time ("dropout-like'')**. Given the selected hidden layer, MasP uses the similarity map derived from the [SEG] token and image tokens as a soft logit mask to prompt SAM for mask generation, **offering explicit spatial cues through its activation regions**.
>
> For simplicity, we provide the UGround data flow pipeline below.
>
> In original work LISA [1],
>
> input: ($x\_{img}$, $x\_{txt}$) $\rightarrow$ LLaVA (layers: $\ell=0,1,..., 31$) $\rightarrow$ **(c)fixed last layer:** $\boldsymbol{h}\_{seg}^{\left( 31 \right)}\in \mathbb{R} ^{4096}$ $\rightarrow$ SAM: $\hat{\mathbf{M}}$
>
> In our UGround, we advance LISA [1] via components **(a): Stochastic Skip Connection (SSC)** and **(b): Mask
> as Prompt (MasP)**:
>
> input: ($x\_{img}$, $x\_{txt}$) $\rightarrow$ LLaVA (layers: $\ell=0,1,..., 31$) $\rightarrow$ **(a) layer select**: e.g., $\ell^*$=18 $\rightarrow$ **(b) MasP**:$\mathcal{M}^{(18)}$, $\boldsymbol{h}\_{seg}^{\left( 18 \right)}\in \mathbb{R} ^{4096}$ $\rightarrow$ SAM: $\hat{\mathbf{M}}$
>
> [1] LISA: Reasoning Segmentation via Large Language Model, CVPR 2024.

---

> > ### Author Response · Authors · 2025-11-19
> > **Rebuttal by Authors**
> >
> > >Which limitations of previous Large Multimodal Models for visual grounding are being addressed, and how does this method solve them?
> >
> > We view the "limitations of previous Large Multimodal Models for ..." at two levels: **method-level** and **architecture-level**.
> >
> > **At the method-level**, we have already explicitly pointed out the limitations **in the abstract (Lines 14–19**):  *“UGround addresses two primary challenges posed by the prevailing paradigm: (1) its reliance on the fixed last hidden layer, which sequentially amplifies cumulative errors arising from layer-by-layer propagation without intermediate correction, and (2) its use of <SEG> as a prompt, which implicitly projects textual embeddings into visual space without explicit spatial cues (e.g., coordinates).”* In the Introduction (Lines 74–94), we further elaborate on these two points.
> >
> > To this end, we propose two key components: 1) **Stochastic Skip Connection(SSC)**. and 2) **Mask as Prompt(MasP)**, as stated in our paper (**Abstract, Lines 18-26; Sec. 1 introduction, Lines 95-107**). SSC is a reinforcement learning policy that, via stochastic sampling, allows each [SEG] token to slide across unrolled transformer layers, enabling dynamic layer selection at which it connects to the vision model (e.g., SAM) **in a skip-connection fashion ("skip-connection-like'')**[2]. By skip subsequent ones of $\ell^*$, we can ''cheat'' this telephone game (Main text: Lines 82-94) by letting the final participant (SAM) tap into intermediate layers (e.g., 1–39) in advance. Across several forward pass, **such mechanism virtually connects all layers to SAM while activating only one pass at a time ("dropout-like'')**. Given the selected hidden layer, MasP uses the similarity map derived from the [SEG] token and image tokens as a soft logit mask to prompt SAM for mask generation, **offering explicit spatial cues through its activation regions**.
> >
> > **At the architecture-level**, we clearly point out the problems that exist in current unified frameworks. In Sec. 2 Related Work, 2) **Unified Segmentation Models.** Depending on the semantic granularity of joint tasks, we divide the literature into two groups: *versatility-oriented segmentation* and *attribute-oriented segmentation*. We, for the first time, unify visual grounding within a single framework from an attribute perspective.  **As stated in our paper (Sec. 2 Related Work, Lines 153–161)**:
> > *"Closest to our work, PixelLM (Ren et al., 2024) cannot tackle empty targets, while GSVA (Xia et al., 2024) fails to handle multi-target reasoning. Attribute-oriented unification is more fundamental, as it can be applied from a versatile perspective to any specialized task. Yet, we observe that little research has looked into unifying existing tasks within a single framework from an attribute perspective, we therefore present UGround."*
> >
> > To this end, we unify multiple complex visual grounding tasks within a single, coherent framework. **This involves issues at the implementation level, such as designing task-specific dataloaders for different datasets, supporting multiple [SEG] tokens for multi-target scenarios, and handling false premises by adding support for empty targets through [REJ] token**. These details can be found in the implementation code provided in the [supplementary material](https://openreview.net/attachment?id=pWi3tvhhmx&name=supplementary_material), and the directory structure of the code is shown in the appendix (Lines 657–681).

---

> > > ### Author Response · Authors · 2025-11-19
> > > **Rebuttal by Authors**
> > >
> > > >The empirical results are close to existing state-of-the-art methods. Why these results are significant enough to justify the architectural modifications?
> > >
> > > We report results on three datasets: ReasonSeg, RefCOCO, and gRefCOCO. **ReasonSeg serves as our main reported dataset**, showing a significant improvement (ReasonSeg test: gIoU $60.3 \rightarrow 63.6$, cIoU $60.0 \rightarrow 65.4$). **RefCOCO and gRefCOCO are used for generalization experiments**. RefCOCO, as a classical dataset, has already been extensively fine-tuned, so the fact that UGround achieves **“marginally better or just competitive performance compared to the current state-of-the-art for this task on RefCOCO/+/g”** is non-trivial. In addition, we use significantly less training data for RefCOCO (~10k images) compared to prior state-of-the-art. Finally, we also observe a notable improvement on the multi-target dataset gRefCOCO (gRefCOCO val: gIoU $66.47 \rightarrow 72.46$, cIoU $63.29 \rightarrow 65.56$, N-acc. $62.43 \rightarrow 74.53$).
> > >
> > > Table 3. **Benchmarking reasoning segmentation models on the ReasonSeg dataset**, sorted in ascending order of cIoU on the test set. *: reproduced from official models. ft: fine-tuned on 239 samples.
> > >
> > > | Method | val gIoU | val cIoU | test short gIoU | test short cIoU | test long gIoU | test long cIoU | test overall gIoU | test overall cIoU |
> > > |------|:--------:|:--------:|:---------------:|:---------------:|:---------------:|:---------------:|:----------------:|:----------------:|
> > > | X-Decoder | 22.6 | 17.9 | 20.4 | 11.6 | 22.2 | 17.5 | 21.7 | 16.3 |
> > > | Grounded-SAM | 26.0 | 14.5 | 17.8 | 10.8 | 22.4 | 18.6 | 21.3 | 16.4 |
> > > | SEEM | 25.5 | 21.2 | 20.1 | 11.5 | 25.6 | 20.8 | 24.3 | 18.7 |
> > > | OVSeg | 28.5 | 18.6 | 18.0 | 15.5 | 28.7 | 22.5 | 26.1 | 20.8 |
> > > | GRES | 22.4 | 19.9 | 17.6 | 15.0 | 22.6 | 23.8 | 21.3 | 22.0 |
> > > | **7B Models** |  |  |  |  |  |  |  |  |
> > > | *SESAME | 40.3 | 41.6 | 28.9 | 26.3 | 37.3 | 31.9 | 34.9 | 30.7 |
> > > | LLaVA1.5-7B + OVSeg | 38.2 | 23.5 | 24.2 | 18.7 | 44.6 | 37.1 | 39.7 | 31.8 |
> > > | *GSVA | 45.6 | 41.5 | 37.9 | 36.5 | 44.3 | 46.0 | 42.8 | 43.8 |
> > > | *PixelLM | 49.7 | 49.6 | 39.5 | 38.8 | 49.5 | 45.6 | 47.1 | 44.3 |
> > > | LISA-7B | 52.9 | 54.0 | 40.6 | 40.6 | 49.4 | 51.0 | 47.3 | 48.4 |
> > > | HyperSeg-3B | 59.2 | 56.7 | - | - | - | - | - | - |
> > > | VISA-7B | 52.7 | 57.8 | - | - | - | - | - | - |
> > > | VideoLISA-3.8B | 61.4 | 67.1 | 43.8 | 42.7 | 56.9 | 57.7 | 53.8 | 54.4 |
> > > | LISA-7B-LLaVA1.5 (ft) | 61.3 | 62.9 | 48.3 | 46.3 | 57.9 | 59.7 | 55.6 | 56.9 |
> > > | READ-7B-LLaVA1.5 (ft) | 59.8 | 67.6 | 52.6 | 49.5 | 60.4 | 61.0 | 58.5 | 58.6 |
> > > | LISA++-7B-LLaVA1.5 (ft) | 64.2 | 68.1 | 49.6 | **51.1** | 59.3 | 61.7 | 57.0 | 59.5 |
> > > | RSVP-GPT | 64.7 | 63.1 | **55.4** | 50.4 | 61.9 | 62.5 | 60.3 | 60.0 |
> > > | UGround-7B-LLaVA1.5 (ft) | **66.1** | **72.1** | 55.1 | 48.5 | **66.3** | **70.2** | **63.6** | **65.4** |
> > > | **13B Models** |  |  |  |  |  |  |  |  |
> > > | LLaVA1.5-13B + OVSeg | 37.9 | 26.4 | 27.1 | 19.4 | 46.1 | 40.6 | 41.5 | 34.1 |
> > > | LISA-13B-LLaVA1.5 | 57.7 | 60.3 | 50.8 | 50.0 | 54.7 | 50.9 | 53.8 | 50.8 |
> > > | LISA-13B-LLaVA1.5 (ft) | 65.0 | 72.9 | 55.4 | 50.6 | 63.2 | 65.3 | 61.3 | 62.2 |
> > > | READ-13B-LLaVA1.5 (ft) | - | - | 55.4 | **53.7** | 64.4 | 65.1 | 62.2 | 62.8 |
> > > | UGround-13B-LLaVA1.5 (ft) | **67.9** | **74.9** | **57.2** | 50.9 | **67.5** | **69.4** | **65.0** | **65.5** |

---

> > > > ### Author Response · Authors · 2025-11-19
> > > > **Rebuttal by Authors**
> > > >
> > > > Table 4. **Benchmarking referring segmentation models on the RefCOCO(+/g) dataset**,  sorted in ascending order of cIoU on the RefCOCOg val set.
> > > >
> > > > | Method | refCOCO val | refCOCO testA | refCOCO testB | refCOCO+ val | refCOCO+ testA | refCOCO+ testB | refCOCOg val(U) | refCOCOg test(U) |
> > > > |------|:-----------:|:-------------:|:-------------:|:------------:|:--------------:|:--------------:|:---------------:|:----------------:|
> > > > | MCN | 62.4 | 64.2 | 59.7 | 50.6 | 55.0 | 44.7 | 49.2 | 49.4 |
> > > > | VLT | 67.5 | 70.5 | 65.2 | 56.3 | 61.0 | 50.1 | 55.0 | 57.7 |
> > > > | CRIS | 70.5 | 73.2 | 66.1 | 62.3 | 68.1 | 53.7 | 59.9 | 60.4 |
> > > > | LAVT | 72.7 | 75.8 | 68.8 | 62.1 | 68.4 | 55.1 | 61.2 | 62.1 |
> > > > | X-Decoder | - | - | - | - | - | - | 64.6 | - |
> > > > | ReLA | 73.8 | 76.5 | 70.2 | 66.0 | 71.0 | 57.7 | 65.0 | 66.0 |
> > > > | SEEM | - | - | - | - | - | - | 65.7 | - |
> > > > | Segment Anyword | 55.3 | 47.9 | 66.0 | 55.6 | 47.4 | 67.0 | 58.4 | 60.1 |
> > > > | VISA-7B | 72.4 | 75.5 | 68.1 | 59.8 | 64.8 | 53.1 | 65.5 | 66.4 |
> > > > | SESAME | 74.7 | - | - | 64.9 | - | - | 66.1 | - |
> > > > | LISA-7B | 74.9 | 79.1 | 72.3 | 65.1 | 70.8 | 58.1 | 67.9 | 70.6 |
> > > > | PixelLM-7B | 73.0 | 76.5 | 68.2 | 66.3 | 71.7 | 58.3 | 69.3 | 70.5 |
> > > > | READ-7B | 78.1 | 80.2 | 73.2 | 68.4 | 73.7 | 60.4 | 70.1 | 71.4 |
> > > > | SegLLM-7B | 80.2 | 81.5 | 75.4 | 70.3 | 73.0 | 62.5 | 72.6 | 73.6 |
> > > > | GSVA-7B | 77.2 | 78.9 | 73.5 | 65.9 | 69.6 | 59.8 | 72.7 | 73.3 |
> > > > | OMG-LLaVA | 78.0 | 80.3 | 74.1 | 69.1 | 73.1 | 63.0 | 72.9 | 72.9 |
> > > > | GLaMM-7B | 79.5 | 83.2 | 76.9 | 72.6 | **78.7** | 64.6 | 74.2 | 74.9 |
> > > > | POPEN-7B | 79.3 | 82.0 | 74.1 | **73.1** | 77.0 | 65.1 | **75.4** | 75.6 |
> > > > | UGround-7B | **80.6** | **83.5** | **77.7** | 72.8 | 77.5 | **65.6** | 74.7 | **76.1** |
> > > >
> > > > Table 5. **Benchmarking generalized referring expression segmentation (GRES) models on the gRefCOCO** dataset, sorted in ascending order of cIoU on the gRefCOCO validation set. Values are taken from~\citep{liu2023gres}.  N-acc.: the accuracy of correct null-target classification. ft: fine-tuned on gRefCOCO training set.
> > > >
> > > > | Method | Validation gIoU | Validation cIoU | Validation N-acc. | Test A gIoU | Test A cIoU | Test A N-acc. | Test B gIoU | Test B cIoU | Test B N-acc. |
> > > > |------|:---------------:|:---------------:|:----------------:|:-----------:|:-----------:|:--------------:|:-----------:|:-----------:|:--------------:|
> > > > | MattNet | 48.24 | 47.51 | 41.15 | 59.30 | 58.66 | 44.04 | 46.14 | 45.33 | 41.32 |
> > > > | LTS | 52.70 | 52.30 | - | 62.64 | 61.87 | - | 50.42 | 49.96 | - |
> > > > | VLT | 52.00 | 52.51 | 47.17 | 63.20 | 62.19 | 48.74 | 50.88 | 50.52 | 47.82 |
> > > > | CRIS | 56.27 | 55.34 | - | 63.42 | 63.82 | - | 51.79 | 51.04 | - |
> > > > | LAVT | 58.40 | 57.64 | 49.32 | 65.90 | 65.32 | 49.25 | 55.83 | 55.04 | 48.46 |
> > > > | ReLA | 63.60 | 62.42 | 56.37 | 70.03 | 69.26 | 59.02 | 61.02 | 59.88 | 58.40 |
> > > > | **7B Models** | | | | | | | | | |
> > > > | LISA-Vicuna-7B | 32.21 | 38.72 | 2.71 | 48.54 | 52.55 | 6.37 | 39.65 | 44.79 | 5.00 |
> > > > | GSVA-Vicuna-7B | 63.32 | 61.70 | 56.45 | 70.11 | 69.23 | 63.50 | 61.34 | 60.26 | 58.42 |
> > > > | LISA-Vicuna-7B (ft) | 61.63 | 61.76 | 54.67 | 66.27 | 68.50 | 50.01 | 58.84 | 60.63 | 51.91 |
> > > > | GSVA-Vicuna-7B (ft) | 66.47 | 63.29 | 62.43 | 71.08 | 69.93 | 65.31 | 62.23 | 60.47 | 60.56 |
> > > > | UGround-LLaVA1.5-7B (ft) | **72.46** | **65.56** | **74.53** | **74.29** | **70.87** | **73.93** | **66.85** | **61.84** | **71.22** |

---

> > > > > ### Author Response · Authors · 2025-11-19
> > > > > **Rebuttal by Authors**
> > > > >
> > > > > >How much of the performance improvement comes from using SAM, and how much comes from the new method itself?
> > > > >
> > > > > **"using SAM" is not our modifications**. Such a paradigm was originally proposed by LISA [1] and has been adopted by its successors, e.g., SESAME [2] and GSVA [3]. SAM is used for mask generation/decoding, and the new method itself cannot generate masks without a decoder. Reviewer can **refer to Table 3, Line 392 and Line 394 for comparison**: LISA-7B-LLaVA1.5 (ft) vs. UGround-7B-LLaVA1.5 (ft) (ReasonSeg test: gIoU $55.6 \rightarrow 63.6$, cIoU $56.9 \rightarrow 65.4$). **Our code is built upon LISA [1], and both models use SAM as the segmentation decoder**. Here, we provide the data flow pipeline between LISA and UGround. We also recommend reviewer refer to LISA [1] for more details.
> > > > >
> > > > > In original work LISA [1],
> > > > >
> > > > > input: ($x\_{img}$, $x\_{txt}$) $\rightarrow$ LLaVA (layers: $\ell=0,1,..., 31$) $\rightarrow$ **(c)fixed last layer:** $\boldsymbol{h}\_{seg}^{\left( 31 \right)}\in \mathbb{R} ^{4096}$ $\rightarrow$ SAM: $\hat{\mathbf{M}}$
> > > > >
> > > > > In our UGround, we advance LISA [1] via components **(a): Stochastic Skip Connection (SSC)** and **(b): Mask
> > > > > as Prompt (MasP)**:
> > > > >
> > > > > input: ($x\_{img}$, $x\_{txt}$) $\rightarrow$ LLaVA (layers: $\ell=0,1,..., 31$) $\rightarrow$ **(a) layer select**: e.g., $\ell^*$=18 $\rightarrow$ **(b) MasP**:$\mathcal{M}^{(18)}$, $\boldsymbol{h}\_{seg}^{\left( 18 \right)}\in \mathbb{R} ^{4096}$ $\rightarrow$ SAM: $\hat{\mathbf{M}}$
> > > > >
> > > > > [1] LISA: Reasoning Segmentation via Large Language Model, CVPR 2024.
> > > > >
> > > > > >Please make clarifications for the mathematical nomenclature.
> > > > >
> > > > > We have fixed the typo in Table 2 (Lines 219 – 234), $\mathcal{S}\_{\mathrm{IoU}}\rightarrow \mathcal{M}\_{\mathrm{IoU}}$.  We have also proofread the entire manuscript to ensure that the upcoming version is more rigorous and easier for readers to understand. Also, we will add a notation table for clarifications of key mathematical nomenclature.

---

> > > > > > ### Author Response · Authors · 2025-11-19
> > > > > > **Official Comment by Authors**
> > > > > >
> > > > > > Dear reviewer, we would like to thank you for your time and efforts. We hope that your concerns are addressed with our rebuttal. Please let us know if there are any further questions that need clarification.

---

> > > > > > > ### Author Response · Authors · 2025-11-26
> > > > > > > **The revised paper is now available.**
> > > > > > >
> > > > > > > Hi Reviewer aGp8, the revised paper is now available at [https://openreview.net/pdf?id=pWi3tvhhmx](https://openreview.net/pdf?id=pWi3tvhhmx). All modifications are clearly highlighted in **blue** throughout the manuscript.
> > > > > > >
> > > > > > > * *"to which refer different variants for using similarity as mask (Line 219 – 234) is not clear."*: We have rewritten Lines 219–234 for clarity.
> > > > > > >
> > > > > > > * *"Please make clarifications for the mathematical nomenclature."*: We have added a notation table in Lines 704-722 for clarity.
> > > > > > >
> > > > > > > The main paper has been carefully checked and polished.

---

### Author Response · Authors · 2025-11-19
**Author Rebuttal by Authors**

We thank all the reviewers for their time and efforts. We are encouraged that the reviewers find that:

- The core idea of "unrolling" transformers and using a learned policy to dynamically select an intermediate layer is highly novel (L9cY), simple and effective (Bxic).
- We have made significant contributions, including
  - The paper is supported by strong and comprehensive empirical evidence(L9cY), good motivation (Bxic).
  - The method significantly outperforms recent SOTA methods on diverse and challenging datasets (ReasonSeg, gRefCOCO) (L9cY).
  - Strong performance: Competitive performance on ReasonSeg, RefCOCO(+/g) and gRefCOCO (Bxic).
  - The work makes a significant contribution by successfully unifying multiple complex visual grounding tasks within a single, coherent framework (L9cY).
  - Honorable mention to the attention given to details (e.g., cleaned and underlined proceedings names in the reference, high-quality figures) reflects a high degree of care (L9cY).
  - The ablation studies are thorough, validating the contribution of each component of the proposed PPM mechanism (dynamic selection, mask as prompt, reward formulation), which substantiates the design choices (L9cY). Adequate ablation: Tables 6, 7, 8 show that the proposed modules and design choices are somewhat effective (Bxic).

- The paper is very well-written and presented. The motivation is clearly articulated through insightful analysis, and the proposed method is explained with clarity and well-designed figures (Figure 1, Figure 3) (L9cY), good soundness and good presentation (Bxic).


We attempted our best to address the questions as time allowed. We believe the revisions have made the paper stronger and thank all the reviewers for their help.

---

### Comment · Area_Chair_YKH9 · 2025-11-23
**Reviewer-Author Discussion**

Hi Reviewers,

Please kinly and actively participate in the review-author dicussion, raise your further concerns so that the authors can explain more, and make your final decisions.

---

### Note · Authors · 2026-01-26

I have read and agree with the venue's withdrawal policy on behalf of myself and my co-authors.

---

### Meta-Review · Area_Chair_X6Pa · 2025-12-25

**Summary:**

From the perspective of contributions, this paper primarily emphasizes two aspects: “unified” and “unrolled.” However, in my view, the “unified” claim is not well substantiated. The so-called unification refers to integrating three previously separate capabilities (a. supporting only referring segmentation or reasoning segmentation; b. detecting only a single object; c. inability to respond to empty targets). Yet, what the authors term “multi-target” actually refers to multi-category rather than multi-instance prediction—i.e., responding with multiple categories in a single dialogue. This is a point of criticism, as multi-category support has been addressed in earlier works such as GeoPixel. Multi-instance segmentation remains a current challenge; while LISA++ has achieved multi-instance segmentation (via Hungarian matching), its performance is still limited. Moreover, LISA++ already accomplishes points (a) and (b), and regarding point (c), although LISA++ does not provide detailed experiments, it is trained on LLaVA data and should theoretically be capable of rejecting empty targets. Therefore, I consider the contribution here to be relatively modest.

Regarding the “unrolled” aspect, the paper presents a more solid contribution. It investigates the impact of extracting features from different hidden layers on performance and develops SSC and MasP based on this exploration. One point of discussion, however, is the necessity of using reinforcement learning (RL) for layer selection. The authors emphasize differentiability as a motivation for using RL, but differentiability does not inherently require RL, e.g., gumbel softmax is another option. Beyond this, the experiments are thorough, and the analysis is generally reasonable and credible. That said, on some datasets, the results only show marginal improvements over the current state-of-the-art.

Taken together, these factors place the paper in the category of a moderate, rather than a standout, contribution to the field.

**Reviewer Concerns:**

Based on the author responses and revisions, here is a summary of reviewer concerns categorized as addressed and outstanding:

**Addressed Concerns**
*   **Clarification of Mathematical Notation**: The authors have clarified the mathematical naming conventions in response to Reviewer aGp8's criticism.

*   **Explication of Contributions**: The authors have provided a more detailed explanation of their stated contributions compared to prior work to Reviewer aGp8.

*   **Inference Cost & RL Complexity**: The authors have satisfactorily addressed the concerns raised by Reviewers L9cY and Bxic regarding the computational overhead of inference and the complexity introduced by the Reinforcement Learning (RL) component.

*   **General Writing and Presentation Issues**: The authors have made revisions in response to general writing quality points raised by Reviewer Bxic.

**Outstanding Concerns**

*   **Novelty and Significance of the Core Contribution:**

"Unified" Architecture Claim (aGp8): Both the reviewer and the AC maintain that the paper's central claim of a "unified" architecture (handling referring/reasoning segmentation, multi-category) represents only a modest advance, as similar capabilities exist in prior works like LISA++ and GeoPixel. The rebuttal did not sufficiently counter this critique, leaving the work appearing more like an engineering integration.

*   **Insufficient Methodological Justification:**

**Necessity of RL (Bxic):** The authors failed to provide experimental proof justifying the use of RL for hard gating. The lack of a comparative baseline (e.g., CE loss with hard gating) weakens the argument for RL's necessity.

**Rationale for Layer Selection Strategy (L9cY):** The authors acknowledged the lack of experimental validation for their layer selection strategy, which depends solely on the [SEG] token's hidden state. The response that this would be explored in a future journal version is considered speculative and does not resolve the methodological concern.

**Exploration of Reward Signals (L9cY):** The authors did not supplement the rebuttal with additional experiments exploring alternative reward signals as suggested.

*   **Manuscript Polish:**

**Persistent Grammatical/Formatting Errors:** Specific issues, such as the capitalization error on line 266, remain in the revised manuscript, indicating incomplete final polishing.

**Reviewer Scores:**

Regarding Reviewer aGp8's comments (initial score: 2):
The primary criticisms concern the paper's writing, the presentation of related work, and the claimed significance of its main contributions. While some grammatical issues persist in the revised manuscript (e.g., capitalization error on line 266), the authors have provided helpful clarifications on mathematical notation in their rebuttal. They also further elaborated on their contributions compared to prior work—namely, the unified architecture and the dynamic layer selection for the [SEG] token. However, I remain skeptical about the novelty of the "unified" contribution, as I believe existing works (e.g., LISA++, GeoPixel) have largely achieved similar capabilities. In this respect, the paper's contribution appears modest and more akin to an engineering integration.
Overall, I consider the authors to have partially addressed Reviewer aGp8’s concerns. This reviewer is likely to maintain a score of 2 or possibly raise it to 4.

Regarding Reviewer L9cY's comments (initial score: 8):
This reviewer focused on inference overhead, the complexity introduced by RL, the image-dependent layer selection strategy, and the exploration of reward signals. The authors adequately addressed the questions on inference cost and RL complexity. However, they acknowledged that in the current UGround model, layer selection relies solely on the hidden state of the [SEG] token, without experimental validation of this design. They noted that this would be explored in a future journal version, which makes their response somewhat speculative. Similarly, no additional experiments were provided regarding alternative reward signals.
Overall, I expect Reviewer L9cY will likely maintain the score of 8.

Regarding Reviewer Bxic's comments (initial score: 4):
This reviewer raised concerns about the lack of clear analysis for dynamic layer selection, the justification for using RL, computational costs, and writing quality. The authors have provided a relatively comprehensive response to most of Bxic’s points. That said, the necessity of RL remains unresolved. The authors argued that CE with soft gating is suboptimal, which is reasonable, but RL is not the only way to implement hard gating. The absence of experiments comparing CE with hard gating weakens their argument.
Overall, I anticipate Reviewer Bxic will likely keep the score at 4.

---

### Decision · Program_Chairs · 2026-01-26

Reject